# Antigen phagocytosis by B cells is required for a potent humoral response

Ana Martínez-Riaño[1], Elena R Bovolenta[1], Pilar Mendoza[1], Clara L Oeste[1], María Jesús Martín-Bermejo[1], Paola Bovolenta[1], Martin Turner[2] (ID), Nuria Martínez-Martín[1,*] (ID) & Balbino Alarcón[1,**] (ID)

## Abstract

**Successful vaccines rely on activating a functional humoral response that results from promoting a proper germinal center (GC) reaction. Key in this process is the activation of follicular B cells that need to acquire antigens and to present them to cognate CD4 T cells. Here, we report that follicular B cells can phagocytose large antigen-coated particles, a process thought to be exclusive of specialized antigen-presenting cells such as macrophages and dendritic cells. We show that antigen phagocytosis by B cells is BCR-driven and mechanistically dependent on the GTPase RhoG. Using $Rhog^{-/-}$ mice, we show that phagocytosis of antigen by B cells is important for the development of a strong GC response and the generation of high-affinity class-switched antibodies. Importantly, we show that the potentiation effect of alum, a common vaccine adjuvant, requires direct phagocytosis of alum–antigen complexes by B cells. These data suggest a new avenue for vaccination approaches by aiming to deliver 1–3 μm size antigen particles to follicular B cells.**

**Keywords** alum; B cells; phagocytosis; vaccination
**Subject Categories** Immunology; Signal Transduction

## Introduction

The humoral response is essential for the efficacy of most vaccines by generating high-affinity immunoglobulins that are able to neutralize the infection before it has spread in an uncontrollable manner. The humoral response begins with the recognition of antigen by cognate B cells leading to B-cell activation and internalization of antigen. Those activated B cells migrate to the interface between the T- and B-cell area [1] where they present the internalized and processed antigen to cognate T cells. At this step, B cells can either differentiate toward antibody-secreting plasmablasts or migrate deep into the follicle to generate a germinal center (GC)

response. GC B cells undergo different rounds of proliferation and sequential interactions with follicular dendritic cells (FDC) and T follicular helper cells (TFH) [2–4]. The GC reaction leads to somatic hypermutation (SHM) and Ig class switch recombination (CSR) followed by the differentiation of B cells either into antibody-producing high-affinity plasma cells or into memory B cells.

The BCR has a dual role serving both as a signaling receptor, inducing B-cell activation, and as a receptor for the selective internalization of antigen. Once internalized, antigens are transported through the endo-lysosomal pathway to be processed into small peptides that are loaded onto MHC class II molecules (MHC-II) and presented to CD4 T cells [3,4]. It is considered that B-cell activation is predominantly elicited by antigens presented on the membrane of antigen-presenting cells (APCs) [5]. Nevertheless, it has also been observed that soluble and small particulate antigens reach the secondary lymphoid organs and are able to activate, and be taken up, by B cells directly, i.e. independently of APCs [6,7]. Accordingly, it seems that differences in the physical properties and molecular characteristics of the antigen might have an impact on how the BCR recognizes and internalizes the antigen [8]. It is for that reason that, in order to promote a long-lasting humoral response, immunization strategies have focused on the study of adjuvants that can promote a more efficient response. Adjuvants are considered as compounds that are able to enhance the magnitude and length of the specific immune response but with minimal lasting effects on their own. They can be classified according to their components or mechanism of action as immunomodulators (such as TLR agonists) and carriers that present the antigen to the immune system in an optimal manner [9].

There are different cellular mechanisms of antigen acquisition that differ according to antigen size and dependence on membrane receptors. These mechanisms are as follows: endocytosis, micropinocytosis, and phagocytosis. The major pathway for antigen acquisition by B cells is thought to be endocytosis (for antigens < 0.2 μm). Phagocytosis is the process of acquiring large particles (> 0.5 μm) and requires an intense remodeling of the actin cytoskeleton. Phagocytosis is generally believed to be carried out by specialized APCs [10], but not by naïve B cells [11,12]. Notwithstanding, it has been described that a specific subpopulation of B

1   Centro de Biología Molecular Severo Ochoa, CSIC–UAM, Madrid, Spain
2   Laboratory of Lymphocyte Signalling and Development, The Babraham Institute, Cambridge, UK
    *Corresponding author. Tel: +34911964721; Fax: +34911964420; E-mail: nmartinez@cbm.csic.es
    **Corresponding author. Tel: +34911964555; Fax: +34911964420; E-mail: balarcon@cbm.csic.es

cells, B1 B cells, is also able to phagocytose particles including bacteria [13–16].

BCR signaling promotes the activation of actin regulators including Rho family GTPases [17]. Moreover, several mutant mouse models of Rho proteins, such as Cdc42, RhoH, and Rac2, present altered humoral responses [18–22]. RhoG is a member of the Rac subfamily of Rho GTPases, with high homology to Rac1 and Cdc42 in their effector domains [23]. RhoG is ubiquitously expressed and it has been implicated in different cellular processes such as engulfment of apoptotic bodies as well as FcyR- and CR3-dependent phagocytosis in macrophages [24–30]. We previously described that RhoG is required for trogocytosis of APC membrane fragments by mature T cells, a mechanism of antigen acquisition that could mimic a frustrated phagocytosis [31]. A first characterization of $Rhog^{-/-}$ mice showed normal T- and B-cell development, as well as a normal humoral response when mice were immunized with soluble antigens [32].

Here, we report that follicular B cells are able to phagocytose antigens through their BCR and initiate a humoral response against particulate antigens. We observe that the common adjuvant alum induces a potent humoral response via the generation of antigen aggregates that are phagocytosed by cognate follicular B cells. Furthermore, RhoG is required for the phagocytosis of antigen by B cells. Indeed, $Rhog^{-/-}$ mice generate a defective humoral response to particulate antigens, but normal response to soluble ones. We therefore present a new fundamental mechanism for antigen acquisition by B cells in order to mount a correct and potent humoral response, which is mediated by the RhoG GTPase.

## Results and Discussion

### B cells phagocytose antigens and present antigen-derived peptides *in vitro* by a RhoG-dependent process

We wondered whether follicular B cells, similarly to B1 B cells [13,15], could directly phagocytose particulate antigens. To do so, we applied a well-established protocol, which is the use of fluorescent latex beads of 1 and 3 μm in diameter combined with confocal microscopy. In this case, we incubated purified naïve follicular B cells with 1 and 3 μm beads that had been previously coated with goat anti-IgM F(ab)'2 antibody. After the incubation at 37°C, cells were stained at 0°C with a fluorescent anti-goat Ig antibody. This allowed us to distinguish between B cells having membrane-attached beads (positive for the anti-goat Ig antibody) and B cells that had completely internalized beads (negative for anti-goat Ig staining). Using this approach, we could clearly determine by confocal microscopy that follicular B cells were able to phagocytose particles of 1 and 3 μm in diameter, presenting the typical rearrangement of the plasma membrane around the particles while remaining negative for the anti-goat Ig staining (Fig 1A). In order to quantify this phagocytic process, we applied the same principle using flow cytometry. Using this method, we could monitor the percentage of B cells with phagocytosed beads according to their negative staining for the anti-goat Ig antibody, as well as the different number of phagocytosed beads, up to 5, according to the stepwise increase in fluorescent intensity in the bead fluorescence channel (Fig 1B). This method allowed us to calculate a phagocytic

index that reflects the percentage of B cells that have phagocytosed beads and the number of phagocytosed beads per cell (Fig 1B). Using this method, we could corroborate that follicular B cells can phagocytose 1 and 3 μm beads by a BCR-specific process actively, since it is blocked at 0°C (Fig EV1A). Furthermore, we showed that B cells incubated at 37°C and permeabilized with detergent became all positive for anti-goat Ig staining, indicating that anti-goat Ig negative B cells had truly phagocytosed the beads (Fig EV1B). The phagocytic ability of follicular B cells had a size limitation since they were basically unable to internalize 10 μm particles (Fig 1C). Furthermore, beads internalization by B cells was inhibited by cytochalasin D and latrunculin A, two inhibitors of the rearrangement of the actin cytoskeleton, and by PP2, an inhibitor of tyrosine kinases of the src family (Fig EV1C), thus suggesting that it is a bona fide phagocytic process triggered by BCR signaling. These data show that, contrary to general belief [11,12,33], naïve B cells are able to phagocytose antigen-coated particles in a BCR-driven process.

Previous work from our group described that RhoG mediates TCR-driven phagocytic antigen uptake in mature T cells [31]. To determine whether RhoG could also mediate the BCR-driven phagocytic process in B cells, we compared the phagocytic ability of purified $Rhog^{-/-}$ B cells with that of WT cells upon incubation with fluorescent 1 and 3 μm beads coated with anti-IgM. Interestingly, $Rhog^{-/-}$ B cells had fourfold less phagocytic activity than WT B cells (Fig 1B). An example of how RhoG is required for phagocytosis of anti-IgM-coated beads is illustrated in Fig 1D where WT B cells are shown to have both attached (positive for anti-goat IgG in red, asterisk) and phagocytosed 3 μm beads (negative to anti-goat IgG and internal to the B-cell plasma membrane marker B220, arrow), whereas RhoG-deficient B cells had only attached beads (Fig 1D). The differences between WT- and RhoG-deficient B cells were detected prior to complete phagocytosis of attached anti-IgM-coated beads: WT B cells formed a phagocytic cup with more intense BCR (IgM) accumulation and actin cytoskeleton rearrangement than $Rhog^{-/-}$ B cells (Fig EV1D). These results showed that follicular B cells can phagocytose beads by a BCR-driven process that requires the RhoG GTPase.

The phagocytic capacity was previously described in teleost fish, amphibian, and human B cells [14,34]. Interestingly, it was described that phagocytic B cells were able to present antigen to T cells and to produce anti-Salmonella IgMs. More recently, the phagocytic capacity of murine peritoneal B1 cells was demonstrated [13,15], nevertheless, the phagocytic capacity of murine B2 cells was found to be negligible [15]. A possible explanation for this is that the latter authors did not use a BCR-triggering phagocytic stimulus. We think that our results extend beyond the latter by showing that naïve follicular B cells are able to phagocytose antigens by a BCR-dependent mechanism and to present them to T cells, promoting the activation of both types of lymphocyte.

In regard to antigen acquisition by B cells, it has been put forward a model, which proposes that B cell extracts antigen using two different mechanisms depending on the stiffness of the APC's membrane [35,36]. Thus, B cells can either pull off antigen from the APC's membrane or mediate the extracellular digestion of the antigen bound to the APC's membrane. We propose the phagocytic capacity of B cells as a third mechanism to allow antigen internalization by B cells. We hypothesize that this antigen phagocytosis

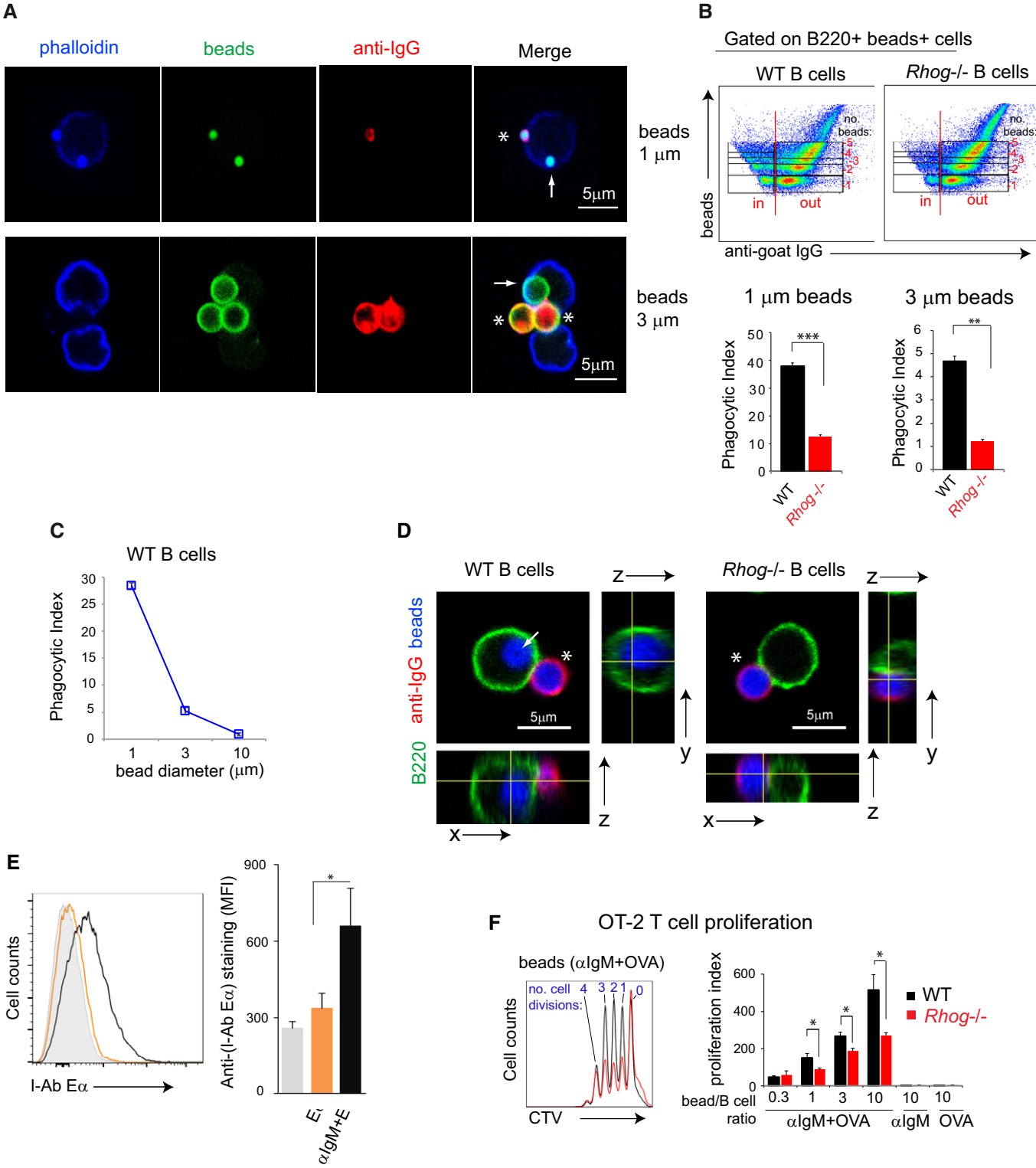

**Figure 1.**

by B cells plays a relevant role when antigens present specific sizes features, for instance, this could be the case for bacteria and yeast.

In T-dependent humoral responses, antigen presentation by B cells is required in order to get CD4 T cell help. To interrogate if B

cells were able to present phagocytosed antigens, we took advantage of the Eα peptide presentation assay [37]. B cells were incubated for 2 h with 1 μm beads coated with Eα peptide alone or Eα peptide plus anti-IgM antibody. Afterward, the abundance of Eα peptide presented by MHC-II on the B-cell surface was quantified by

**Figure 1.  Follicular B cells phagocytose particulates antigens *in vitro* through a RhoG-dependent mechanism.**

A   Confocal section of follicular B cells in the process of phagocytosing 1 and 3 μm beads coated with anti-IgM. Purified follicular B cells were incubated with 1 or 3 μm fluorescent beads coated with a goat anti-mouse anti-IgM for 1 h at 37°C and afterward stained with an anti-goat 488 antibody on ice to distinguish cells with attached or already internalized beads. Beads are shown in green, the extracellular staining with anti-goat IgG in red, and the cortical actin cytoskeleton in blue. Completely phagocytosed beads, negative for anti-goat IgG, are indicated with an arrow, and non-phagocytosed beads are indicated with an asterisk.

B   Flow cytometry plots of WT- and RhoG-deficient B cells incubated for 1 h with 1 μm fluorescent beads coated with anti-IgM antibody and stained afterward extracellularly with anti-goat 488, as in (A). The phagocytic index was calculated according to the stepwise increase in the beads' mean fluorescence intensity and lack of anti-goat 488 staining on B cells with beads. The graphs below the plots show the phagocytic index of WT and *Rhog*$^{-/-}$ B cells incubated with 1 or 3 μm beads. Data represent means ± SEM (*n* = 3).

C   Phagocytic index for WT B cells incubated for 1 h with 1, 3, and 10 μm beads coated with anti-IgM. Data represent means ± SEM (*n* = 3).

D   Confocal section and orthogonal images of follicular WT and *Rhog*$^{-/-}$ B cells in the process of phagocytosing 3 μm beads coated with anti-IgM as in (A). Beads are shown in blue, the extracellular staining with anti-goat IgG in red, and B220 in green. Completely phagocytosed beads, negative for anti-goat IgG, are indicated with an arrow, and non-phagocytosed beads are indicated with an asterisk.

E   Antigen presentation of Eα peptide on MHC-II measured by flow cytometry. Splenic follicular B cells from WT mice were incubated with beads coated with Eα peptide (orange), with Eα + anti-IgM (black), or uncoated (gray) for 2 h. The bar graph shows the means ± SEM of Eα-MHC-II MFI (*n* = 3).

F   Proliferation profiles of OT2 T cells after 3 days of culture with WT (black) or *Rhog*$^{-/-}$ (red) B cells stimulated with 1 μm beads coated with anti-IgM and ovalbumin. The right bar graph shows the OT2 proliferation index upon different anti-IgM+ovalbumin-coated bead:B-cell ratios. As controls, beads were incubated only with anti-IgM or with ovalbumin. Data represent means ± SEM (*n* = 3).

Data information: *\*P* < 0.05; *\*\*P* < 0.005; *\*\*\*P* < 0.0005 (unpaired Student's *t*-test).

flow cytometry using an antibody specific for the Eα peptide bound to I-A$^b$. We confirmed that follicular B cells are able to present antigen peptides bound to MHC-II after antigen phagocytosis (Fig 1E). To determine if phagocytosed antigen was presented to T cells, purified WT and *Rhog*$^{-/-}$ B cells were incubated with different bead/B-cell ratios of 1 μm beads coated with anti-IgM plus ovalbumin and subsequently were cultured with purified Cell Trace Violet (CTV)-labeled CD4 T cells from an OT2 transgenic mouse (Fig 1F). OT2 T cells carry a specific TCR capable of recognizing an ovalbumin-derived peptide presented by I-A$^b$ [38]. After 3 days, OT2 T-cell proliferation was monitored according to the dilution of CTV by flow cytometry. Compared to OT2 T cells incubated with WT B cells, OT2 T cells cultured with *Rhog*$^{-/-}$ B cells proliferated less at all bead/B-cell ratios tested (Fig 1F).

We next interrogated if the defective antigen presentation to T cells was due to the inability of RhoG-deficient B cells to take up antigen by phagocytosis or if it was consequence of a more general defect. To do so, we decided to use B1-8$^{hi}$ mice, whose B cells recognize 4-Hydroxy-3-nitrophenylacetyl hapten (NP) and its derivate 4-Hydroxy-3-iodo-5-nitrophenylacetyl hapten (NIP) with high affinity [39]. We found that B1-8$^{hi}$ B cells were defective in the presentation of antigen to OT2 T cells when NIP-derivatized ovalbumin was given bound to beads but not when given in solution (Fig EV2). Altogether, these *in vitro* results reveal the ability of follicular B cells

to phagocytose and present particulate antigens to cognate T cells by a BCR- and RhoG-dependent process. In order to test if this phagocytic process could play a role in the immune response, we wondered if antigen phagocytosis by follicular B cells could also take place *in vivo*.

### B lymphocytes phagocytose particulate antigens *in vivo*

After describing the novel phagocytic ability of follicular B cells *in vitro*, we moved on to study if antigen phagocytosis by follicular B cells could also take place *in vivo*. WT mice were inoculated intraperitoneally (i.p.) with 1 μm fluorescent beads coated with NIP-OVA. After 5 h, whole spleens were explanted, fixed, and examined using confocal microscopy. Numerous beads were detected in the spleen, both in follicular and in extrafollicular areas (Fig 2A, left). Remarkably, some of the beads were found inside B cells in follicular areas (Fig 2A, right), indicating that beads can reach this compartment in a short period of time. This fact raises a still unanswered question: how these large particulate antigens reach the follicular area and if specific APCs are required. We find these points potentially interesting to study in the future.

To monitor phagocytosis *in vivo* in a quantitative manner, WT and *Rhog*$^{-/-}$ B1-8$^{hi}$ mice were inoculated i.p. with 1 μm fluorescent beads coated with NIP-OVA. After 5 h, cells were isolated from the

**Figure 2.  Antigen-specific splenic B cells phagocytose antigens *in vivo* through a BCR-driven process also dependent on RhoG.**

A   Confocal microscopy images obtained with a 10×, 25×, and 63× objective from spleen sections of WT mice 5 h post-intraperitoneal (ip) immunization with 1 μm fluorescent beads coated with NIP-OVA. The left images (10× objective) show beads (gray) distribution along a spleen slice where MOMA1 staining (red) determines the outer/inside part of the follicles and DAPI (blue) the nucleus. Upper right image shows both the marginal zone (MZ) and follicular area (FO): DAPI (blue); MOMA1 (green); 1 μm bead (gray). The arrow points to a bead located in the FO zone. The below image shows an amplification of a follicular B cell with a 1 μm bead: B220 (red); 1 μm bead (gray).

B   Phagocytosis of 1 μm fluorescent beads covalently bound to NIP-OVA by splenic B cells from WT or *Rhog*$^{-/-}$ B1-8$^{hi}$ mice was assessed after 5 h post-IP immunization through extracellular staining with an anti-ovalbumin antibody. Cytometry plots show the identification of NP-reactive B cells (CD19$^+$ B220$^+$) with attached or intracellular beads (NP$^+$ fluorescent beads$^+$). Anti-OVA staining distinguishes between the attached and the already phagocytosed beads. Quantification charts represent the means ± SEM of phagocytic NP-reactive (NP$^+$) and non-reactive (NP$^-$) B cells in WT (black) and *Rhog*$^{-/-}$ (red) mice.

C   Follicular (CD21$^+$ CD23$^+$) and MZ (CD21$^+$ CD23$^-$) B cells were identified in WT and *Rhog*$^{-/-}$ mice immunized as in (A), and their B-cell phagocytic ability was measured also by anti-OVA staining (CD19$^+$ beads$^+$ anti-Ova$^-$). The bar graph shows the percentage of phagocytic follicular and MZ B cells in WT B1-8$^{hi}$, *Rhog*$^{-/-}$ B1-8$^{hi}$, and non-transgenic WT (non-Tg) mice as a control. Data represent means ± SEM (*n* = 3).

Data information: *\*P* < 0.05 (unpaired Student's *t*-test).

spleen and incubated at 0°C with an anti-ovalbumin antibody that enables to distinguish between cells with external or phagocytized beads (Fig EV3A). Flow cytometry analysis showed that NP-specific

B cells (B220[+] CD19[+] NP[+]) contained beads completely phagocytosed, negative for anti-ovalbumin staining (Fig 2B). Importantly, it was observed that non-reactive NP-negative B cells

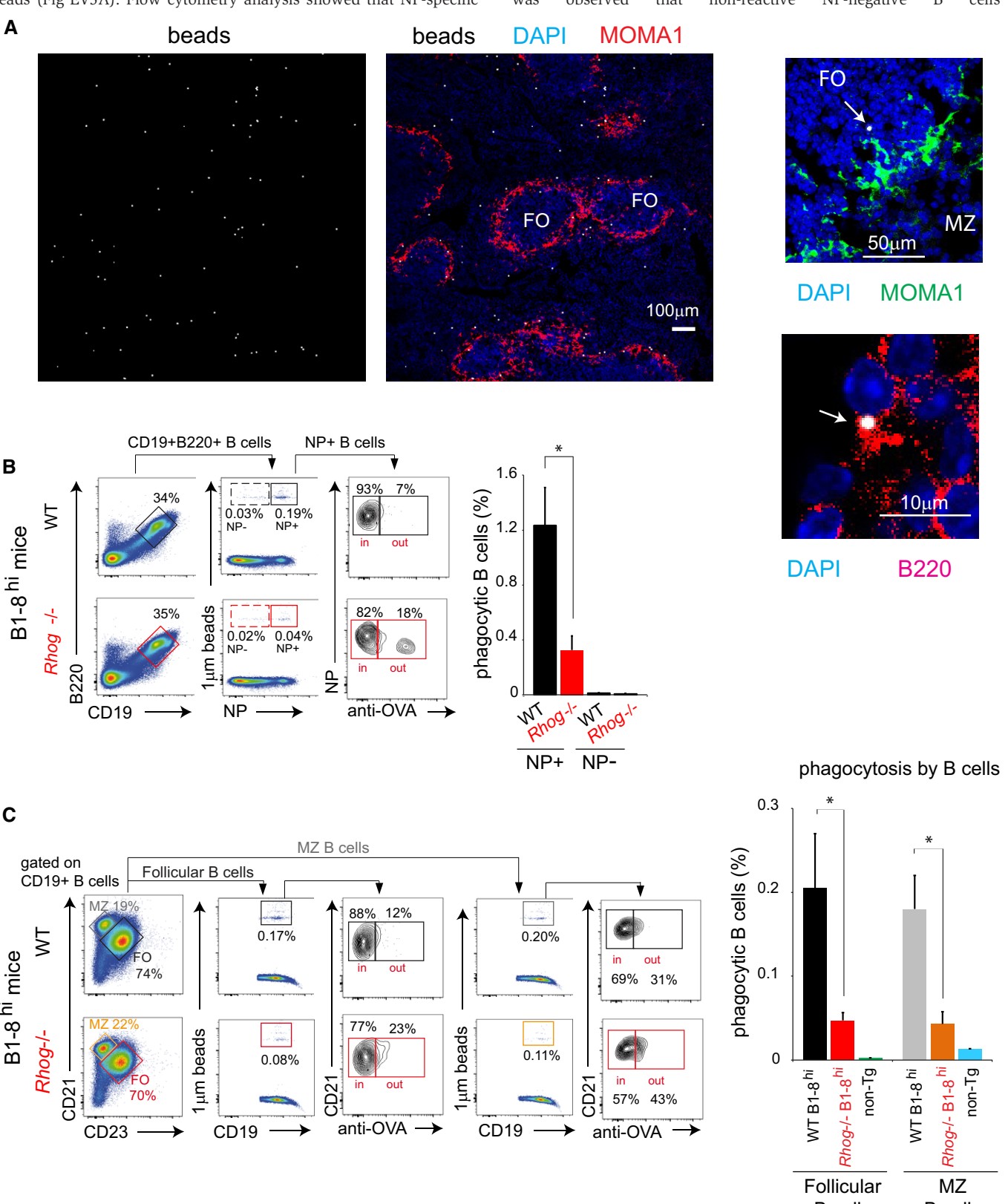

**Figure 2.**

(B220$^+$ CD19$^+$ NP$^-$) were practically unable to phagocytose, corroborating that antigen phagocytosis is driven by BCR engagement (bar plot, Fig 2B). Moreover, $Rhog^{-/-}$ B cells also showed lower phagocytic ability *in vivo*, phagocytosing at least 3-fold less beads than their WT counterparts, thus matching the phagocytic impairment described in the *in vitro* assays (Fig 1B). Using different surface markers, we could compare the phagocytic ability of follicular and marginal zone B cells and found that both were able to phagocytose beads through their BCR to a similar extent (Fig 2C). Remarkably, either follicular or marginal zone B cells seem to require RhoG to phagocytose antigens. Interestingly, we could also monitor this phagocytic activity in other cell types such as macrophages (Fig EV3B). These cells presented a phagocytic ability similar to that of follicular B cells, indicating that BCR-driven phagocytosis by B cells is not negligible (Fig EV3B).

## RhoG is essential to mount an immune response against particulate antigens

Antigen acquisition by B cells and its presentation to CD4 T cells is an essential process that occurs during GC formation. For that reason, we decided to evaluate whether antigen phagocytosis by B cells could play a role in the development of the GC response and therefore in the humoral T-cell-dependent immune response. To do so, 1 μm beads bound covalently to antigen (NIP-OVA) were used as immunogens for i.p. administration. After 5 days of immunization, spleens from immunized WT mice were explanted, fixed, and examined using confocal microscopy. Interestingly, we could observe the presence of an incipient GC reaction (note the GL7$^+$ IgD$^-$ cells), but also that some beads were still present in the B-cell area, close to the GC zone (Fig 3A). This result suggested that the immunogen used was clearly able to reach and remain in the B-cell area, where it could potentially unleash a T-cell-dependent immune response. However, GC B cells were not found to contain phagocytosed beads (Fig 3A). A possible interpretation for this finding is that GC B cells detected 5 days after inoculation of beads are the result of earlier phagocytic events that results in B-cell proliferation and bead dilution. In order to evaluate the role of antigen phagocytosis in the GC response, WT and $Rhog^{-/-}$ mice were immunized i.p. with NIP-OVA bound to 1 μm beads. The response was analyzed using flow cytometry according to the appearance of GC B cells (B220$^+$ CD95$^+$ GL7$^+$) in the spleen 7 days after inoculation (Fig 3B). $Rhog^{-/-}$ mice generated at least 3-fold less GC B cells than their WT counterparts, suggesting that phagocytosis by B cells

is essential to mount a strong GC reaction. In order to study whether the RhoG requirement for the GC response was B-cell intrinsic, purified follicular B cells from WT and $Rhog^{-/-}$ B1-8$^{hi}$ mice bearing the Ly5.2 allele (CD45.2) were transferred into receptor mice bearing the Ly5.1 allele (CD45.1). The latter were subsequently immunized i.p. with NIP-OVA bound to 1 μm beads. Using the congenic marker CD45.2, we were able to distinguish the transferred from the receptor B cells (Fig 3C). Interestingly, although there were similar percentage of transferred B cells after immunization, $Rhog^{-/-}$ B cells showed a strong impairment in the GC reaction, presenting tenfold less GC cells (B220$^+$ CD95$^+$ GL7$^+$) than the WT (Fig 3C). A similar B-cell intrinsic defect in GC formation in the absence of RhoG was detected in response to immunization with bigger, 3 μm beads (Fig 3D). Interestingly, inoculation of $Rhog^{-/-}$ mice with NIP-OVA bound to 1 μm or 3 μm beads resulted in reduced production of IgM and class-switched high-affinity IgG1 specific for NIP compared to WT mice, although the differences were only significant for the bigger beads (Fig 3E). This difference depending of the bead size could be explained by two effects: the more stringent conditions in terms of RhoG requirement for the phagocytosis of bigger particles, such as shown in Fig 1B, and an effect of bigger particles resulting in more antibody production in WT genotype (compare scales in high-affinity IgM and IgG1 plots, Fig 3E). Therefore, based on *in vitro* and *in vivo* results (Figs 2 and 3), it can be proposed that antigen phagocytosis by B cells is crucial to start an effective high-affinity humoral response.

It has been described that antigen internalization in B cells is followed by fusion and polarization of antigen-containing vesicles, being this process essential for efficient antigen presentation, autophagosome maturation, and asymmetrical cell division in B cells [18,40,41]. Based on our experimental results, we propose that impairment on phagocytosis rate is translated into less T-cell activation *in vitro* and consequently a defective GC response. Whether $Rhog^{-/-}$ B cells have also a defect on antigen polarity that could aggravate the already described defect is something that will require future work.

## Alum-based immunizations induce a potent humoral response requiring antigen acquisition by B cells through RhoG-dependent phagocytosis

Mice deficient in RhoG did not show any impairment in T- and B-cell development or in the T-dependent humoral response to a

---

**Figure 3. Antigen phagocytosis by B cells is important for the germinal center response.**

A   Confocal image of spleen sections of WT mice 5 days post-immunization with NIP-OVA bound to 1 μm fluorescent beads. IgD (red); GL7 (green); 1 μm fluorescent beads (gray). White arrows point FO B cells (IgD$^+$) with beads. Representative image of 3 GCs per spleen section and per immunized mouse (n = 6 mice).

B   Analysis of germinal center B cells (CD95$^+$ GL7$^+$) in WT and $Rhog^{-/-}$ mice 7 days post-immunization with 1 μm beads covalently coated with NIP-OVA. The bar graph shows the mean percentage ± SEM of CD95$^+$ GL7$^+$ B cells (n = 3). **P < 0.005 (unpaired Student's t-test).

C   WT and $Rhog^{-/-}$ B1-8$^{hi}$ CD45.2 B cells were adoptively transferred to congenic CD45.1 receptor mice immunized with 1 μm beads covalently bound to NIP-OVA. The flow cytometry panel illustrates germinal center B cells (CD95$^+$ GL7$^+$) within the transferred WT (upper panel) or $Rhog^{-/-}$ (lower panel) B cells (CD45.2$^+$ CD45.1$^-$). Quantification charts show the percentage of transferred B cells (CD45.2$^+$ B220$^+$) and GC B cells (CD95$^+$ GL7$^+$). Data represent means ± SEM (n = 3). **P < 0.005 (unpaired Student's t-test).

D   Quantification chart of the percentage of germinal center B cells within the WT and $Rhog^{-/-}$ B1-8$^{hi}$ CD45.2 B cells adoptively transferred to congenic CD45.1 receptor mice, as in (C), and immunized with 3 μm beads covalently bound to NIP-OVA. Data represent means ± SEM (n = 6). *P < 0.05 (unpaired Student's t-test).

E   WT and $Rhog^{-/-}$ mice were immunized with 1 and 3 μm beads covalently bound with NIP-OVA. Sera were collected after 14 days and high-affinity NP(7) and low-affinity NP(41)-specific IgM (upper graphs) and IgG1 (lower graphs) were measured by ELISA. Graphs show means ± SEM (n = 4) as well as the ratios of absorbance for NP(7) vs. NP(41) binding. n.s. P > 0.05; *P < 0.05; **P < 0.005; ***P < 0.0005; ****P < 0.00005 (unpaired Student's t-test).

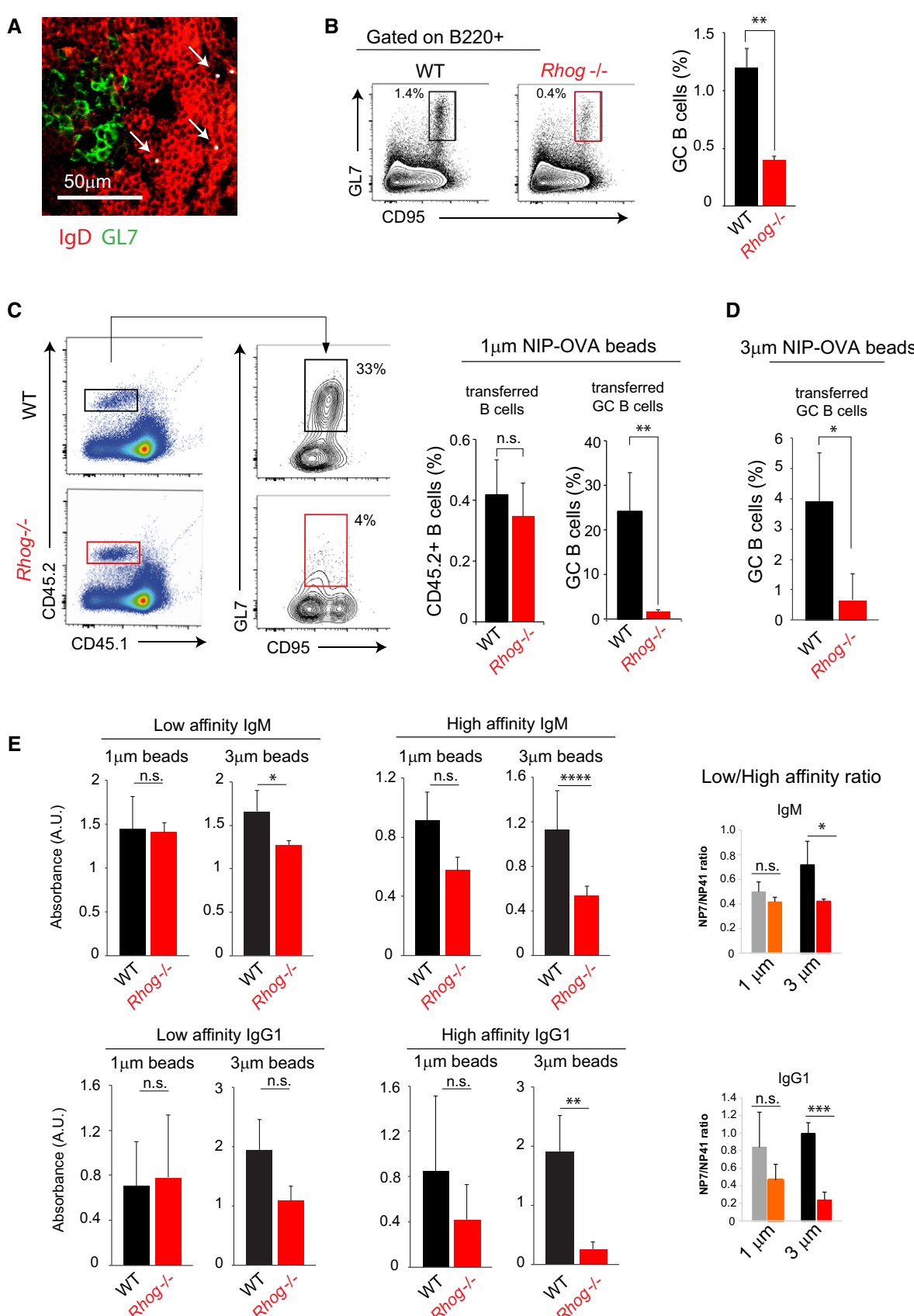

**Figure 3.**

soluble antigen. However, $Rhog^{-/-}$ B cells are deficient in the GC response to phagocytosed beads (Fig 3B). Since the use of beads coated with antigens is not yet a feasible method for immunization, we decided to analyze the role of B-cell antigen phagocytosis in the humoral response using a common vaccination protocol that could induce antigen phagocytosis by B cells. In this regard, it has been described that the adjuvant alum promotes a potent humoral response due to the generation of antigen aggregates with a size ranging from 2 to 10 µm. The antigen/alum aggregates have been shown to promote their phagocytic uptake by macrophages and DCs [42]. Therefore, we tested whether antigen/alum complexes could also be directly phagocytosed by B cells. To do so, purified follicular B cells from B1-8$^{hi}$ mice were incubated for 1 h with a preparation of alum with NIP-BSA-FITC (fluoresceinated alum, see Materials and Methods). By confocal microscopy, we observed that NIP-BSA-FITC complexed with alum could be detected forming particles with a different range of sizes (1 to 7 µm; Fig 4A). Strikingly, we could observe B cells that managed to phagocytose these antigen aggregates whereas $Rhog^{-/-}$ B cells did not (Fig 4B). A quantification of antigen–alum complex phagocytosis was carried out by flow cytometry using NIP-BSA-FITC plus alum conjugates and an anti-FITC antibody to distinguish internal alum complexes from adherent ones (Fig 4C). $Rhog^{-/-}$ B cells were strongly deficient in the uptake of alum–antigen complexes. These results suggested that antigen phagocytosis by B cells could be playing a role in a broad array of immunizations where alum is used as an adjuvant.

We next assessed the role of antigen phagocytosis by B cells and, at the same time, re-evaluated the role of RhoG in mounting an effective immune response against particulate antigens. We therefore immunized WT and $Rhog^{-/-}$ mice with two different types of immunogen: NIP-OVA complexed with alum (particulate antigen) and NIP-OVA diluted in PBS with or without LPS (soluble antigen). The presence of NP-specific antibodies in the serum of these mice was measured by ELISA 7 and 14 days after immunization. The response in terms of high-affinity IgG1 was stronger for WT mice immunized with NIP-OVA plus alum complexes than in those immunized with soluble NIP-OVA plus LPS (Fig 5A). Interestingly, while $Rhog^{-/-}$ mice did not present deficiencies in the antibody response against soluble antigens, i.e. NIP-OVA diluted in PBS plus LPS, they were deficient in the generation of high-affinity NP-specific IgM [NP (7)] in response to NIP-OVA/alum complexes at day 14 after immunization (Fig 5A). This defect in $Rhog^{-/-}$ mice was also observed when low and high-affinity NP-specific IgG1 antibodies were measured 7 and 14 days after immunization (Fig 5A). These results, together with the differences in the ratio high/low-affinity antibodies (Fig 5A), suggest that antigen phagocytosis by B cells plays a fundamental role in the generation of high-affinity antibodies in response to particulate antigens such as alum-based immunizations.

Although immunization with soluble antigen supplemented with LPS had not shown any defect in the absence of RhoG, we still decided to study if the reduced antibody response of $Rhog^{-/-}$ B cells could be due to defective TLR signaling. To do so, we monitored plasma cell differentiation and proliferation of $Rhog^{-/-}$ B cells upon CpG and LPS stimulation in vitro. We found that $Rhog^{-/-}$ B cells proliferated and differentiated into plasma cells (IgD$^-$ CD138$^+$) similarly to, or even slightly more than, their WT counterparts (Fig EV4A and B), indicating that the impaired B-cell response to antigen/alum aggregates is not ancillary to a defect in TLR

activation. In contrast to the response to soluble anti-IgM, the proliferative response of B cells to bead-bound anti-IgM was strongly impaired in the absence of RhoG (Fig EV4B and C). Indicating that the absence of RhoG does not affect the proliferation–differentiation pathway. Strikingly, and only using particulate antigen, we have been able to unmask the until now underestimate role of RhoG in B-cell activation and consequently in the humoral response. Therefore, RhoG seems to be positioned in a very precise molecular crossroad controlling specifically antigen phagocytosis.

To obtain a more detailed understanding of the implications of antigen phagocytosis by B cells in the antibody response to particulate antigens, we analyzed GC formation against particulate and soluble antigens after immunization (NIP-OVA plus alum or NIP-OVA with or without LPS). Compared to the WT, $Rhog^{-/-}$ mice showed normal GC formation after immunization with soluble NIP-OVA, with or without LPS, according to the appearance of GC cells (B220$^+$CD95$^+$GL7$^+$) in spleens 7 days after immunization (Fig 5B and C). By contrast, $Rhog^{-/-}$ mice showed a reduction of at least 50% in the response to NIP-OVA plus alum (Fig 5B and C). In order to study whether these defects were B-cell intrinsic, we performed an adoptive transfer experiment in which CD45.2$^+$ B cells from WT B1-8$^{hi}$ and $Rhog^{-/-}$ B1-8$^{hi}$ were transferred to WT CD45.1$^+$ mice. Once transferred, receptor mice were immunized with NIP-OVA plus alum. In this context, $Rhog^{-/-}$ transferred B cells showed a defective GC response, reduced NIP-reactive B-cell expansion, and poor class switch to IgG1 7 days after immunization (Fig 5D).

Altogether, these data indicate that RhoG-dependent antigen phagocytosis by B cells is fundamental for the generation of the humoral response against particulate antigens. Moreover, the mechanism of action of alum to boost the humoral immune response is based, at least in part, on the intrinsic ability of B cells to phagocytose. These results justify the need to include from now on the concept of soluble vs. particulate antigen to evaluate the role of any new molecule in the humoral response, especially those ones related to actin cytoskeleton remodeling. Moreover, the fact that the lack of RhoG does not completely abrogate the phagocytic ability of B cells suggests that other proteins could have a redundant function. We consider compulsory to re-evaluate the role of other molecules in the humoral immune response. This could be the case for Cdc42 or Rac, which also have a role controlling the actin cytoskeleton, and have been described to modulate the humoral response [18,21,43,44].

More importantly, our work opens a new window in vaccination research; the description of the phagocytic capacity of B cells takes us to think that it could be potentially useful to design strategies to prime this as-yet unrevealed B-cell feature using different adjuvants such as alum. We do think that new vaccine strategies should try to promote the delivery of antigens in a particulate manner to follicular B cells in order to generate a potent humoral response.

Recently, it has been described a novel in vitro platform to produce antigen-specific human antibodies based on the use of particles (0.11 µm) coated with anti-κ together with CpG to co-stimulate at the same time BCR and TLR9 receptors, respectively. Authors claim that particle internalization allows CpG to reach TLR9 intracellular compartment, something that is not possible in a soluble manner [45]. This particle internalization results in the synergistic activation of both receptors inducing a more robust B-cell proliferation and differentiation into plasma cells

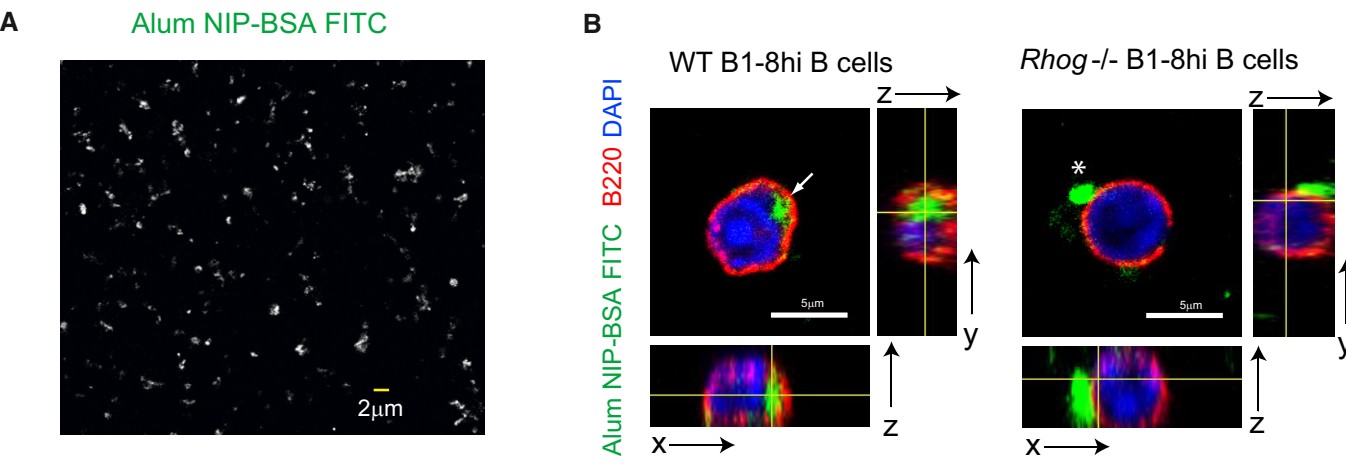

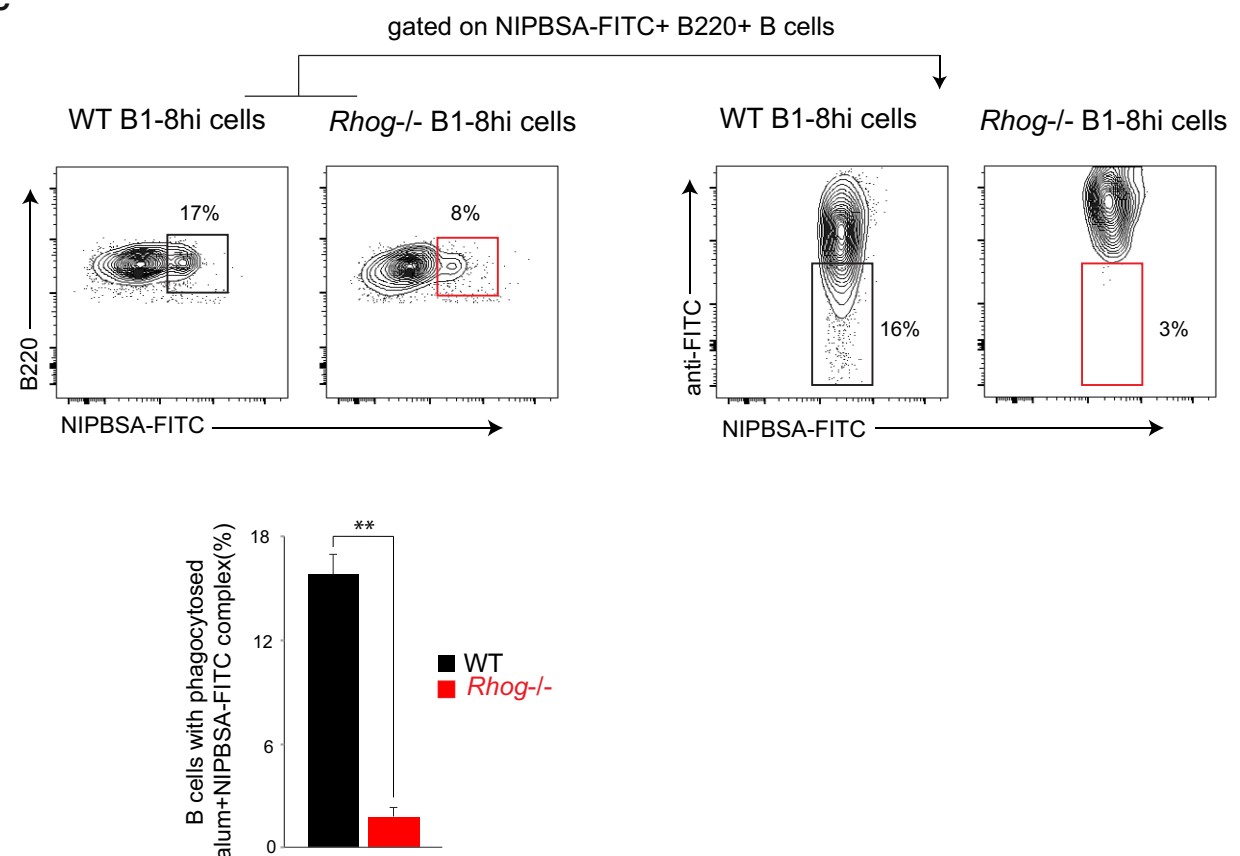

**Figure 4.  Alum–antigen aggregates are phagocytosed by B cells.**

A   Confocal microscopy image of antigen aggregates generated with NIP-BSA-FITC complexed with alum.

B   Confocal microscopy image and orthogonal views of WT and *Rhog*[−/−] B1-8[hi] B cells after 2 h of incubation at 37°C with antigen aggregates generated with NIP-BSA-FITC complexed with alum. B220 (red), DAPI (blue), and NIP-BSA-FITC (green). Completely phagocytosed alum aggregate is indicated with an arrow, and non-phagocytosed aggregate is indicated with an asterisk.

C   Flow cytometry plots of WT- and RhoG-deficient B1-8[hi] B cells incubated for 2 h with antigen aggregates generated with NIP-BSA-FITC complexed with alum and stained afterward extracellularly with an anti-FITC 647 antibody to distinguish those B cells with only internalized aggregates from those still attached to the membrane. B cells positive for NIP-BSA-FITC aggregates were analyzed for lack of anti-FITC staining. The graph below the plots shows the percentage of WT (black) and *Rhog*[−/−] (red) B cells with phagocytosed NIP-BSA-FICT aggregates. Data represent means ± SEM (*n* = 3). **P < 0.005 (unpaired Student's *t*-test).

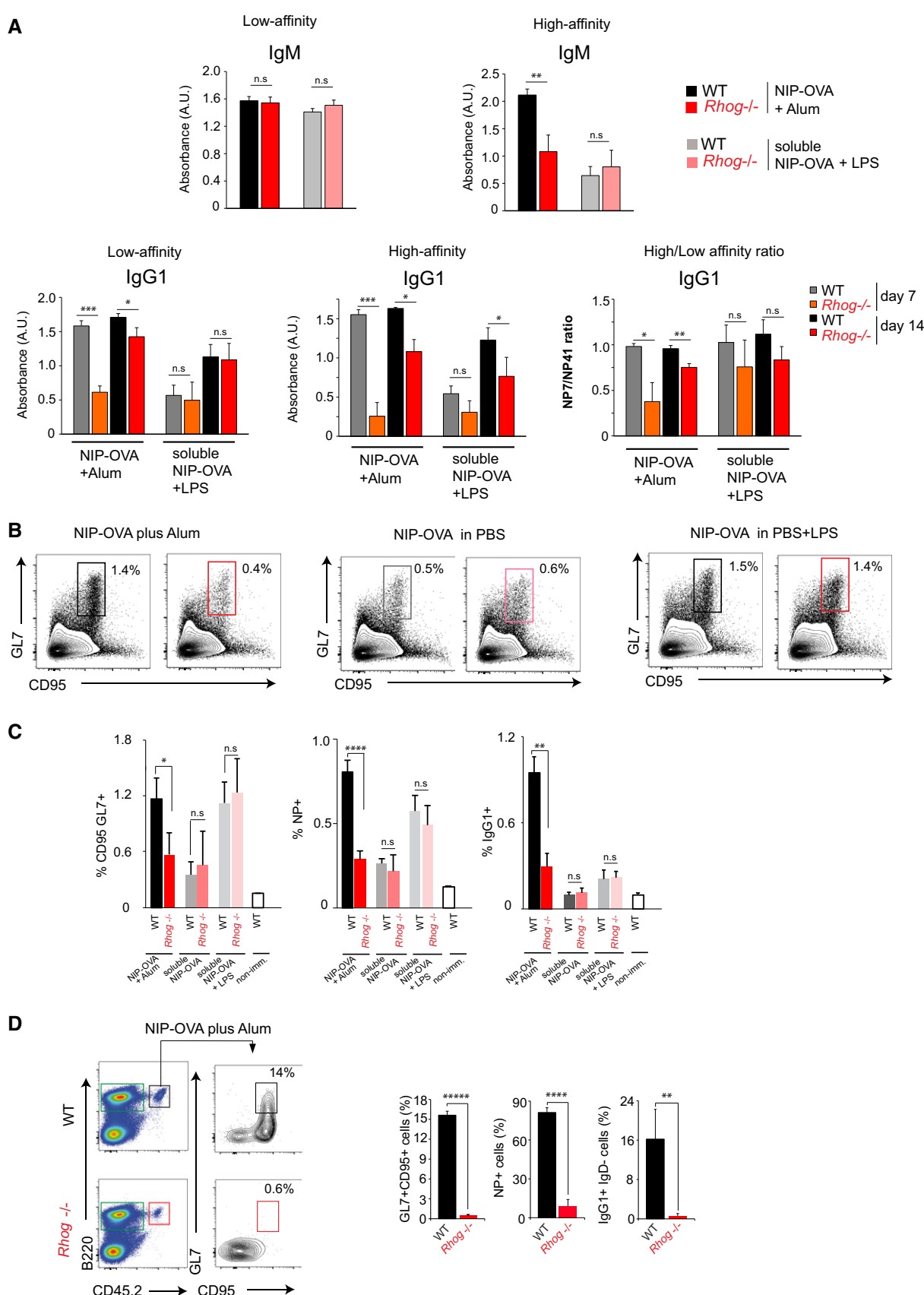

**Figure 5.**

**Figure 5. Alum-based vaccination induces a potent humoral response dependent on RhoG GTPase.**

A  WT and *Rhog*$^{-/-}$ mice were immunized with NIP-OVA complexed with alum or NIP-OVA diluted in PBS and supplemented with LPS. Sera from WT and *Rhog*$^{-/-}$ mice were collected at different time points (7 and 14 days) and high-affinity NP(7) and low-affinity NP(41)-specific IgM and IgG1 were measured by ELISA. IgM data are shown only at day 14 after immunization. Graphs show means ± SEM (n = 4) as well as the ratios of absorbance for NP(7) vs. NP(41) binding.

B  Analysis of germinal center B cells (CD95$^+$ GL7$^+$) on WT (black; gray) and *Rhog*$^{-/-}$ (red; pink) mice 7 days post-immunization with NIP-OVA complexed with alum, NIP-OVA diluted in PBS, or NIP-OVA diluted in PBS and supplemented with 50 μg of LPS.

C  Quantification chart of the mean ± SEM of the percentage of GC (CD95$^+$ GL7$^+$), NP$^+$-specific B cells (NP$^+$ B220$^+$), and IgG1 class-switched (IgG1$^+$ IgD$^-$) B cells of mice immunized as in (B) (n = 5).

D  WT and *Rhog*$^{-/-}$ B1-8$^{hi}$ CD45.2 purified B cells were adoptively transferred to CD45.1 receptor mice immunized with NIP-OVA complexed with alum. Flow cytometry plots show germinal center (CD95$^+$ GL7$^+$) B cells in the transferred B cells (B220$^+$ CD45.2$^+$). Quantification charts represent the mean ± SEM of the percentage of GC (GL7$^+$CD95$^+$), NP-specific (NP$^+$ B220$^+$), and IgG1 class-switched (IgG1$^+$ IgD$^-$) B cells (n = 3).

Data information: *$P < 0.05$; **$P < 0.005$; ***$P < 0.0005$; ****$P < 0.00005$; *****$P < 0.000005$ (unpaired Student's *t*-test).

*in vitro* than when soluble stimulus is used. It is interesting to see that once again the use of particles unmasks a new B-cell feature. It would be interesting to see whether the activation of phagocytosis in B cells is behind this boost of plasma cell differentiation *in vitro*.

Overall, we propose that phagocytosis by B cells is a key step to allow an efficient humoral response against particulate antigens. Furthermore, by unmasking this B-cell feature, we have not only contributed to improve the knowledge on the mechanism of action of the widely used adjuvant alum, also we set the path to potentiate this B-cell intrinsic function and improve future vaccine strategies.

## Materials and Methods

### Mice

*Rhog*$^{-/-}$ mice have been previously described [32]. These mice were crossed with NP-specific B1-8$^{hi}$ knock-in mice bearing a pre-rearranged V region [46]. Mice transgenic for the OT-2 TCR specific for peptide 323-339 of chicken ovalbumin presented by I-A$^b$ [38] and C57BL/6 bearing the pan-leukocyte marker allele CD45.1 were kindly provided by Dr. Carlos Ardavín (CNB, Madrid). All animals were backcrossed to the C57BL/6 background for at least 10 generations. For all *in vivo* experiments, age (6–10 weeks) and sex were matched between the *Rhog*$^{+/+}$ (WT) and *Rhog*$^{-/-}$ mice. Mice were maintained under SPF conditions in the animal facility of the Centro de Biología Molecular Severo Ochoa in accordance with pertinent national and European guidelines. All animal procedures were approved by the ethical committee of the Centro de Biología Molecular Severo Ochoa.

### Antibodies and reagents

The following antibodies were used: anti-mouse CD45R-FITC -V450 –biotin –APC (RA3-6B2), CD4 -PerCP (RM4-5), CD8 -biotin (53-6.7), CD11b -biotin (M1/70), purified CD16/32 (2,4G2), CD19 -PE-Cy7 (1D3), CD25 -APC (3C7), CD43 -biotin (S7), CD45.1 -APC-Cy7 (A20), CD45.2 -APC (104), CD95 –FITC –PE-Cy7 (Jo2), CD138 –APC (3C7), Gr1-biotin (RB6-8C5), GL7 –647 (GL7), IgD –FITC –V450 (11.26c), NK1.1 –biotin from BD Pharmingen; anti-mouse F4/80 –biotin (BM8) and anti-kappa –biotin (RMK-12), eBioY-Ae (Y-Ae from eBiosciences; anti-goat IgGs -FITC –647 and anti-IgM F(ab')2 from Jackson Immunoresearch; anti-Ovalbumin FITC and MOMA-1 from Abcam.

Fluoresbrite Carboxy microspheres in different sizes and colors from Polysciences were used: 1 μm Y/G (15702), 3 μm Y/G (17147), 10 μm Y/G (18142), and 3 μm Y/O (19393). Unlabeled Polybead Carboxylate-Modified Microspheres 1 μm (08226-15) are from Polysciences. FluoSpheres Carboxylate-Modified 1 μm Crimson (F8816) are from Invitrogen. Cell Trace Violet (C34557) is from Life Technology. DAPI (268298) is from Merck. Ovalbumin (A5503) and PH-TRICT (P-1951) are from Sigma. NIP(15)-Fluorescein-BSA (N-5040F-10), NP(7)-BSA (N5050L-10), NP(41)-BSA (N-5050H-10), and NP(36)-PE (N-5070-1) are from Biosearch Technology. Imject Alum (77161) is from Thermo Scientific. Mouse recombinant IL-4 (214-14) is from Peprotech. Mouse recombinant IL-5 (405-ML-005) is from R&D. SBA Clonotyping system-HRP (5300-04) is from Southern Biotech. CpG OD (1826) is from Invivogen. LPS (L2630) is from Sigma.

### Cell preparation and purification

The lymph nodes and spleen from 6- to 8-week-old mice were homogenized with 40 μm strainers and washed in phosphate-buffered saline (PBS) containing 2% (vol/vol) fetal bovine serum (FBS). Spleen cells were resuspended for 3 min in AcK buffer (0.15 M NH$_4$Cl, 10 mM KHCO$_3$, 0.1 mM EDTA, pH 7.2–7.4) to lyse erythrocytes and washed in PBS 2% FBS.

For culture and *in vitro* assays, B cells from spleens were negatively selected using a combination of biotinylated anti-CD43 and anti-CD11b antibodies and incubation with streptavidin beads (Dynabeads Invitrogen) for 30 min and separated using Dynal Invitrogen Beads Separator. B1-8$^{hi}$ B cells were purified using biotinylated anti-CD43, anti-CD11b, and anti-kappa antibodies. OT2 T cells from lymph nodes and spleen were purified using a mix of biotinylated antibodies: anti-B220, anti-CD8, anti-NK1.1, anti-CD11b, anti-GR1, and anti-F4/80. Splenic and lymph node B and T cells were maintained in RPMI 10% FBS supplemented with 2 mM L-glutamine, 100 U/ml penicillin, 100 U/ml streptomycin, 20 μM β-mercaptoethanol, and 10 mM sodium pyruvate.

### Flow cytometry

Mouse single-cell splenocyte suspensions were incubated with fluorescence-labeled antibodies for 30 min at 4°C after blocking FC receptors using an anti-CD16/32 antibody. Afterward, cells were washed in PBS + 1% BSA and data were collected on a FACS Canto II. Analysis was performed using FlowJo software.

### Antigen-coated bead preparation

To prepare beads with adsorbed antigen, a total of $130 \times 10^6$ carboxylated latex beads 1 μm in diameter were incubated overnight with a concentration of 40 μg/ml of protein in 1 ml of PBS at 4°C. For preparation of antigen-coated beads 3 and 10 μm in diameter, bead concentration was reduced gradually; 3-fold and 30-fold, respectively. Beads were subsequently washed twice with PBS plus 1% BSA and resuspended in RPMI medium. To prepare beads with covalently bound antigen, the PolyLink Protein Coupling Kit (Polysciences) was used as specified by the manufacturer. An equivalent of 12.5 mg of beads was washed in Coupling Buffer (50 mM MES, pH 5.2, 0.05% Proclin 300), centrifuged 10 min at 1,000 *g*, and resuspended in 170 μl Coupling Buffer. A 20 μl volume of Carbodiimide solution (freshly prepared at 200 mg/ml) was added to the bead suspension and incubated for 15 min. After that, a total of 400 μg of NIP-OVA was added at a final concentration of 5 mg/ml. Incubations were carried out at room temperature with gentle mixing. Beads were centrifuged and washed twice in Wash/Storage buffer (10 mM Tris, pH 8.0, 0.05% BSA, 0.05% Proclin 300). To remove non-covalent bound protein, beads were washed once with 0.1% SDS followed by two washes with PBS + 1% BSA for SDS removal.

### Antigen bead-bound phagocytosis assays

Naïve follicular WT or *Rhog*$^{-/-}$ B cells were resuspended in RPMI containing 20 mM Hepes plus 0.2% BSA and plated in 96-well V-bottom plates at a concentration of $1 \times 10^6$ cells in 50 μl. Antibody-coated florescent beads were added to reach a bead:cell ratio of 10 (1 μm beads), 3 (3 μm beads), or 1 (10 μm beads). The cell and bead suspensions were briefly centrifuged at 400 *g* and were incubated at 37°C for different time points. Subsequently, cells were washed and stained on ice with a fluorescent isotype-specific Ig antibody to track the presence of beads bound to the cells that had not been phagocytosed. At this stage, the cells were either analyzed by flow cytometry (FACS Canto II) or incubated for 15 min on coverslips coated with poly-L-lysine and then processed for immunofluorescence.

The phagocytic index was calculated according to the stepwise increase in bead MFI, corresponding to the number of beads that the B cell contains, and lack of anti-goat FITC antibody staining. Phagocytic index = (1 bead inside*1) + (2 beads inside*2) + (3 beads inside*3) + …

When actin and Src-inhibitors were used, B lymphocytes were pre-treated for 1 h with cytochalasin D (1 μg/ml), latrunculin A (20 μg/ml), or PP2 (20 μM) and incubated afterward with 1 and 3 μm beads coated with anti-IgM. The inhibitors treatment was maintained during the incubation with the beads.

### Alum phagocytosis assay

For alum aggregate phagocytosis, NIP-BSA-FITC aggregates in alum were generated incubating 1 mg/ml NIP-BSA-FITC 1:1 with alum adjuvant for 30 min at RT with mild shaking and washed 3× afterward. $0.5 \times 10^6$ purified follicular B1-8$^{hi}$ B cells from spleen WT or *Rhog*$^{-/-}$ B1-8$^{hi}$ mice were purified and incubated with NIP-BSA-FITC complexed with alum for 2 h at 37°C. Subsequently, cells were washed and stained on ice with an anti-FITC 647 antibody to track the presence of NIP-BSA-FICT complexes attached to the B cells that had not been phagocytosed. At this stage, the cells were either analyzed by flow cytometry (FACS Canto II) or incubated for 15 min on coverslips coated with poly-L-lysine and then processed for immunofluorescence.

### Eα peptide presentation

$0.5 \times 10^6$ purified B cells were incubated for 2 h at 37°C with only 1 μm beads or with 1 μm beads (10:1 beads:B-cell ratio) covered with Eα peptide (20 μg/ml) or Eα peptide (20 μg/ml) + anti-IgM (20 μg/ml). Afterward, cells were washed and stained with anti-MHC-II/Eα antibody and analyzed by FACS.

### *In vivo* phagocytosis assay

B1-8$^{hi}$ mice were immunized intraperitoneally with $2 \times 10^7$ Crimson fluorescent beads 1 μm in diameter covalently bound to NIP-OVA. Spleens were harvested after 5 h and were disrupted in PBS + 2% FBS on ice. Cell suspensions were stained with antibodies to identify macrophages (CD11b and F4/80), B cells (CD19 and B220), and marginal zone and follicular B cells (CD23 and CD21). To identify phagocytosed beads from those just attached to the membrane, cells were stained with anti-Ovalbumin-FITC 1:100 dilution for 30 min. Samples were analyzed by flow cytometry (FACS Canto II). All *ex vivo* procedures were performed at 0°C.

When spleens were used for confocal microscopy, they were fixed in paraformaldehyde (PFA) 4% for 24 h, left in 30% sucrose for 24 h more, and embedded in OCT.

### Adoptive transfer and immunizations

To assess the formation of GC B cells *in vivo* and the generation of anti-NP antibodies, mice were immunized i.p. with 200 μg of soluble NIP-OVA with or without 50 μg LPS in 200 μl of PBS. Alternatively, mice were immunized with 200 μg of NIP-OVA complexes with 100 μl of alum diluted 1:1 with PBS. For immunization with NIP-OVA bound to beads, a total of $20 \times 10^6$ 1 μm beads covalently bound to NIP-OVA were administered i.p. in 200 μl PBS. After 7 days post-immunization, spleens were harvested and analyzed by flow cytometry to detect germinal center cells. When 1 and 3 μm beads were compared, $70 \times 10^6$ of 1 μm and $20 \times 10^6$ of 3 μm were used. When NP antibodies were studied, animals were bled at 0, 7, and 14 days after immunization.

For adoptive transfer into CD45.1 mice, $1 \times 10^7$ purified B cells from spleens of B1-8$^{hi}$ WT and *Rhog*$^{-/-}$ were injected intravenously. Recipient mice were immunized intraperitoneally with 200 μg NIP-OVA complexed with alum or with $2 \times 10^7$ 1 μm beads bound covalently to NIP-OVA.

### Proliferation and stimulation assays

Proliferation of B cells was assessed using Cell Trace Violet (CTV) labeling as specified by the manufacturer (Thermofisher). A total of $2 \times 10^5$ purified naïve B cells were CTV-stained and cultured for 3 days with LPS (1 μg/ml), CpG (1 μg/ml), anti-IgM (3 μg/ml) + CpG (1 μg/ml), anti-IgM (3, 10, 30 μg/ml) or 1 μm bead-bound anti-IgM (4:1, 10:1, 15:1 ratio beads:B cells) in RPMI

supplemented with IL4 (2.5 ng/ml) and IL5 (5 ng/ml). After 3 days, PC markers (CD138$^+$ IgD$^-$) and CTV-dilution were analyzed by flow cytometry (FACS Canto II) and FlowJo software.

### Antigen presentation assay

Proliferation of OT2 was assessed with purified CFSE-stained OT2 T cells at a 1:1 ratio with purified B cells together with antigen-coated beads in a round-bottom 96-well plate. For the bead-bound stimulus, B cells were pre-incubated with 1 μm beads coated with anti-IgM plus ovalbumin at different bead:B-cell ratios. After 3–4 days of culture, cells were washed in PBS + 1% BSA analyzed by FACS (FACS Canto II) and FlowJo software.

When presentation of soluble vs. bead-bound antigen was compared, proliferation of OT2 was assessed with purified CTV-stained OT2 T cells at a ratio 1:1 with purified B cells from WT or $Rhog^{-/-}$ B1-8$^{hi}$ mice. For the bead-bound stimulus, 1 mm beads coated with NIP-OVA protein at 3:1 bead:B-cell ratio were used, while for soluble stimulus, 100 ng/ml of soluble NIP-OVA was used.

### Capping and actin-reorganization measurement

$0.5 \times 10^6$ purified follicular B cells from WT or $Rhog^{-/-}$ mice were incubated for 1 h at 37°C with 3 μm beads (1:1 ratio beads: B cell) coated with anti-IgM antibody in a round-bottom 96-well plate. During the last 15 min of incubation, cells were transferred to poly-L-lysine-treated coverslips. Afterward, cells were fixed in 4% PFA for 20 min, washed in PBS + 1% BSA, and stained with anti-IgM 647 in PBS + 1% BSA for 30 min. Subsequently, actin cytoskeleton staining with phalloidin diluted in PBS + 5% FBS + 0.3% Triton X-100 was performed for 1 h at RT. Confocal images were acquired with a Zeiss LSM710 system and a Zeiss AxioObserver LSM710 Confocal microscope.

### Measurement of antigen-specific antibodies

In immunized mice, sera were obtained 7 and 14 days later. Plate-bound NP(7)-BSA and NP(41)-BSA (5 μg/ml) were used to measure high- and low-affinity immunoglobulins in 1:175 dilutions of sera from immunized mice. SBA Clonotyping System-HRP (Southern Biotech) was used to perform the ELISA. Absorbance at 405 nm was determined with an iMark Microplate Absorbance Reader (Bio-Rad).

### Confocal microscopy

For alum aggregate phagocytosis, $0.5 \times 10^6$ purified follicular WT or $Rhog^{-/-}$ B1-8$^{hi}$ B cells were incubated with NIP-BSA-FITC complexed with alum for 2 h at 37°C. Afterward, cells were washed in PBS, fixed in 4% PFA for 20 min, and transferred to poly-L-lysine-treated coverslips. Cells were stained for biotinylated B220 in PBS + 1% BSA for 30 min, washed 2× and stained with Strepta-vidin-TRICT for 15 min. After that, cells were stained with DAPI for 5 min. Confocal images were acquired with a Zeiss LSM710 system and a Zeiss AxioObserver LSM710 Confocal microscope.

For the bead phagocytosis assay, after staining with an anti-goat Alexa488 (for Crissom of Y/O florescent beads) or anti-goat

Alexa647 (for Y/G florescent beads) on ice for 30 min, cells were transferred to poly-L-lysine-treated coverslips and fixed in 4% PFA for 20 min. Afterward, anti-B220 staining in PBS + 1% BSA for 30 min or actin cytoskeleton staining with phalloidin diluted in PBS + 5% FBS + 0.3% Triton X-100 was performed for 1 h at RT. Confocal images were acquired with a Zeiss LSM710 system and a Zeiss AxioObserver LSM710 Confocal microscope.

For immunohistologies, spleens were embedded in OCT, frozen in dry ice and 10-μm-wide frozen sections were cut with a cryostat. Sections were blocked and permeabilized with PBS 0.3% Triton X-100 3% BSA (IF blocking buffer) for 1 h. Stainings were performed in blocking buffer for 4 h with a combination of the following antibodies: MOMA1, IgD, B220.

Images were analyzed using ImageJ software. In order to make 1 μm beads visible in the low magnification spleen images at 5 h post-immunization, beads image was processed as a binary image to dilate the beads pixels.

### Quantification and statistical analysis

Statistical parameters including the exact value of $n$, the means $\pm$ SD, or SEM are reported in the figures and figure legends. A non-parametric two-tailed unpaired $t$-test was used to assess the confidence intervals.

**Expanded View** for this article is available online.

### Acknowledgements

We thank all laboratory members for critical reading of the manuscript. We are also indebted to Cristina Prieto, Valentina Blanco, and Tania Gómez for their expert technical assistance. This work was supported by grants SAF2013-47975-R and SAF2016-76394-R (to B.A.) from the CICYT, by grant from the European Research Council ERC 2013-Advanced Grant 334763 "NOVARIPP" (to B.A.), and from the Fundación Ramón Areces (to the CBMSO). MT is funded by the Biotechnology and Biological Sciences Research Council.

### Author contributions

AM-R designed and performed research, analyzed the data, and wrote the manuscript; ERB and PM assisted with analysis of the humoral response *in vivo*; CLO edited the manuscript; MJM-B and PB helped in immunohisto-chemistry-related experiments; MT provided $Rhog^{-/-}$ mice; NM-M wrote the manuscript and BA supervised and designed research, analyzed the data, and wrote the manuscript.

### Conflict of interest

The authors declare that they have no conflict of interest.

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
