## [Review Process File · EMBO Reports]

Antigen phagocytosis by B cells is required for a potent humoral response

Ana Martínez-Riaño, Elena R. Bovolenta, Pilar Mendoza, Clara L. Oeste, María Jesús Martín-Bermejo, Paola Bovolenta, Martin Turner, Núria Martínez-Martín and Balbino Alarcón

Review timeline:

Submission date:	27 February 2018
Editorial Decision:	22 March 2018
Revision received:	17 May 2018
Editorial Decision:	5 June 2018
Revision received:	15 June 2018
Accepted:	21 June 2018

Editor: Achim Breiling

Transaction Report:

1st Editorial Decision

22 March 2018

Thank you for the submission of your research manuscript to EMBO reports. We have now received the reports from the referees that were asked to evaluate your study, which can be found at the end of this email.

As you will see, all referees think the manuscript is of interest, but requires major revision to allow publication in EMBO reports. Referees #2 and #3 have a number of concerns and/or suggestions to improve the manuscript, which we ask you to address in a revised manuscript. As the reports are below, I will not detail them here. We feel, however, that in particular the points by referee #3 need to be addressed experimentally.

Given the constructive referee comments, we would like to invite you to revise your manuscript with the understanding that all referee concerns must be addressed in the revised manuscript and in a point-by-point response. Acceptance of your manuscript will depend on a positive outcome of a second round of review. It is EMBO reports policy to allow a single round of revision only and acceptance or rejection of the manuscript will therefore depend on the completeness of your responses included in the next, final version of the manuscript.

Revised manuscripts should be submitted within three months of a request for revision; they will otherwise be treated as new submissions. Please contact us if a 3-months time frame is not sufficient for the revisions so that we can discuss the revisions further.

Supplementary/additional data: The Expanded View format, which will be displayed in the main HTML of the paper in a collapsible format, has replaced the Supplementary information. You can submit up to 5 images as Expanded View. Please follow the nomenclature Figure EV1, Figure EV2 etc. The figure legend for these should be included in the main manuscript document file in a section called Expanded View Figure Legends after the main Figure Legends section. Additional Supplementary material should be supplied as a single pdf labeled Appendix. The Appendix

includes a table of content on the first page, all figures and their legends. Please follow the nomenclature Appendix Figure Sx throughout the text and also label the figures according to this nomenclature.

For more details please refer to our guide to authors:

<http://embor.embopress.org/authorguide#manuscriptpreparation>

See also our guide for figure preparation:

http://www.embopress.org/sites/default/files/EMBOPress_Figure_Guidelines_061115.pdf

Important: All materials and methods should be included in the main manuscript file.

Regarding data quantification and statistics, can you please specify, where applicable, the number "n" for how many independent experiments (biological replicates) were performed, the bars and error bars (e.g. SEM, SD) and the test used to calculate p-values in the respective figure legends. Please provide statistical testing where applicable. See:

<http://embor.embopress.org/authorguide#statisticalanalysis>

Please also follow our guidelines for the use of living organisms, and the respective reporting guidelines: <http://embor.embopress.org/authorguide#livingorganisms>

We now strongly encourage the publication of original source data with the aim of making primary data more accessible and transparent to the reader. The source data will be published in a separate source data file online along with the accepted manuscript and will be linked to the relevant figure. If you would like to use this opportunity, please submit the source data (for example scans of entire gels or blots, data points of graphs in an excel sheet, additional images, etc.) of your key experiments together with the revised manuscript. Please include size markers for scans of entire gels, label the scans with figure and panel number, and send one PDF file per figure or per figure panel.

- a complete author checklist, which you can download from our author guidelines (<http://embor.embopress.org/authorguide#revision>). Please insert page numbers in the checklist to indicate where the requested information can be found.
- a letter detailing your responses to the referee comments in Word format (.doc)
- a Microsoft Word file (.doc) of the revised manuscript text
- editable TIFF or EPS-formatted single figure files in high resolution (for main figures and EV figures)

Please also note that we now mandate that all corresponding authors list an ORCID digital identifier that is linked to their EMBO reports account!

I look forward to seeing a revised version of your manuscript when it is ready. Please let me know if you have questions or comments regarding the revision.

REFeree REPORTS

Referee #1:

Secondary immune responses with high affinity antibodies rely on germinal center reactions during which follicular B cells interact with T follicular helper cells to acquire somatic mutations in their gene segments that encode the variable domains of Ig heavy and Ig light chains. This manuscript provides solid evidence for three major findings. First, follicular B cells are capable of capturing large particulate antigen complexes. Second, this process is initiated by cognate BCR antigen recognition and requires the small GTPase RhoG. Finally, this kind of antigen uptake drives affinity maturation and Ig class-switching. The presented in vitro and in vivo experiments fully support the authors' conclusion. Also, the manuscript is well written. Collectively, the results significantly contribute to our understanding of potent humoral immunity. Future experiments are necessary to

elucidate the underlying signal mechanism in a more comprehensive manner. At this stage, however, I do not miss additional experiments that will push the manuscript to another level. Hence, I recommend publication as is (which I rarely do).

Referee #2:

This is an interesting paper demonstrating a new role for Rhog in antibody responses. Previously, no significant role for Rhog in antibody responses was found upon immunisation with soluble NP-KLH protein. However, the authors show here that B cell expression of Rhog is important in responses to particulate antigens, including antigens precipitated in alum. In the absence of Rhog in B cells, responses to particulate antigens are impaired in B cell germinal centre formation, class switching, and early production of high affinity antibodies. The authors explain this by showing that antigen-coated particles of 1-3 microns are phagocytosed by B cells and that Rhog is required for the phagocytic uptake and for efficient presentation of the antigens associated with the particles.

Previously, it was thought that uptake of particles that are too big for endocytosis (>200 nm) cannot be taken up by most B cells, except for specialised B cell subsets. However, some of these particles did not contain antigens specific for the BCR. In contrast, there were previous reports showing that the BCR can mediate phagocytic uptake of antigen-coated particles. The data here is novel in that they clarify the role of BCR-driven phagocytosis in B cells and identify Rhog as a component of this pathway that is important for antigen uptake and presentation.

The data is mostly of high quality and in my opinion support the main conclusions. I have however questions about some of the experimental details and also on how some of the data is interpreted.

1. The phagocytosis by B cells is a bit misleadingly analysed in the literature, particularly with respect of the particles that are used for the experiments (latex beads coated or uncoated, bacteria etc). This lead to the early conclusions that only B1 B cells or B cells in certain species can phagocytose. I suggest a more comprehensive discussion and comparison of the published experiments to explain why the author's experiments differ from others. The authors should also cite Souwer et al, *J. Immunol*, 2009, who show BCR-mediated phagocytosis in Ramos B cells, which agrees with their data that B cells are capable of phagocytosis of antigen-coated particles.

2. In Fig. 1, the phagocytic index needs better explanation. The provided formula seems to sum cell numbers with phagocytosed beads, which does not fit with the numbers in the graphs. Are the numbers normalized to total numbers of B cells with beads?

3. In Fig 1D the legend mentions Rhog^{-/-} cells, but the figure only shows one set of data, presumably WT.

4. In Fig 1E, it would be important to know that the absence Rhog does not affect general antigen presentation to T cells, for example using a soluble peptide control.

5. Fig 3A shows presence of beads in follicles, but not in the GC. Is this a reproducible finding? If yes, it suggests that GC B cells do not acquire the antigen from the beads, but from other sources. It would be interesting if the authors could discuss what the source of the antigen is. This could be some soluble antigen contamination in the immunisation prep, or release of antigen from the beads *in vivo*. The release can happen by extracellular degradation, or perhaps after phagocytosis by B cells or other cells. This is important for the interpretation of the GC experiments, and for effects on antibody production, as it suggests that Rhog dependent phagocytosis is important only during the initiation of the response. Such a conclusion is supported by the delay, rather than absence, of antibody production in Rhog^{-/-} mice in the subsequent experiments. Also, the text indicates the analysis is 5 days post immunisation, the figure indicates day 7.

6. In Fig 4B, the authors measure separately high and low affinity antibodies by using low and high valency of the antigen, but technically, the multivalent antigen is going to bind both low and high affinity antibodies. A separate measure of "avidity" determined from the ratio of the two measurements is better. What day post immunisation were the IgM antibodies measured?

7. Also in Fig 4, the overall titers of the IgG1 antibodies are different between the two antigen formulation. This should be mentioned in the interpretation.

8. In Supplementary Fig 3B,C, the authors conclude that the Rhog^{-/-} cells respond normally to IgM and CpG stimulation. The more relevant stimulus, however, is particulate antigen. When the B cells cannot phagocytose the antigen coated beads, do they proliferate or survive normally? This is important to judge whether BCR signalling or other cell physiology in Rhog^{-/-} B cells is normal in the context of particulate antigens.

9. In the discussion, the authors should refrain from concluding that they "prove that phagocytic activity of B cells is key to explain the potent adjuvant effect of alum". The presented data is not sufficient to support this conclusion. Alum does many things and the data in this manuscript do not show how the stimulation of B cell phagocytosis compares to other effects of alum. The data only show that for antigen mixed with alum, B cell expression of Rhog, and therefore B cell phagocytosis, is important.

10. In the abstract and discussion, a range of 1-10 microns is mentioned as a desirable size of particles for immunisation. However, the B cells could not phagocytose the 10 micron beads. 1-3 microns is a more realistic range.

Referee #3:

Report on Manuscript: Antigen Phagocytosis by B cells Required for a potent Humoral Response
In this manuscript the authors address an interesting and relevant topic, the mechanisms used by B cells to internalize large particulate antigens and its importance to generate efficient humoral responses. In particular, they show that follicular B cells are able to phagocytose large antigen-coated particles through their BCR and that this is a fundamental property, which enables efficient humoral responses to such antigens. The authors provide in vitro and in vivo evidence that BCR-mediated phagocytosis by follicular B cells exists and that this process is dependent on the GTPase RhoG. The role of BCR-dependent phagocytosis in humoral responses is further supported by results showing that RhoG knockout mice have defective humoral responses to particulate antigens (1µm), whereas responses elicited by soluble antigens do not seem to be affected. Altogether, these findings are very interesting, however the idea that the BCR mediates phagocytosis of particulate antigens is not original, per se, and has been previously addressed by Batista and Neuberger (Embo J, 2000). The novelty of this work resides on the role of RhoG in mediating phagocytosis of particulate antigens by B cells and its role in promoting humoral responses in vivo. The results presented here are appealing to both the communities in cell biology and immunology. Overall, I recommend this paper for publication in this journal, however I feel certain points should be addressed:

Major points:

Figure 1: Follicular B cells are able to phagocytose particulate antigens. The authors show 2 experimental approaches to demonstrate the phagocytic capacity of follicular B cells. In panel A, immunofluorescent labeling of actin (phalloidin) and anti-IgG (bead) are shown. Beads that are negative for anti-IgG staining are considered to be completely internalized and thus are "protected" from labeling. How do the authors distinguish between antigen degradation on the bead, which would also give negative anti-IgG staining and phagocytosis? The authors should include a set of permeabilized cells to show that the beads remain positive for IgG, under these conditions. Although phagocytosed beads are clearly distinguished from those that are only bound to the cell surface, the experiment would also benefit by staining plasma membrane (and not just actin), where engulfment of beads could be better appreciated.

The data presented suggests that phagocytosis relies on engagement of the BCR and RhoG, however this report should include more information to address this point and not only functional readouts of the antibody responses. For instance, regarding the images, do the authors observe capping of the BCR in B cells interacting with antigen-coated beads? Do Rhog^{-/-} B cells show only surface-bound beads and how is actin organization affected in these cells? Images addressing these questions should be presented in this panel.

Next, the phagocytic process is quantified by flow cytometry (B-C). The results are convincing, however the authors should include the effect of drugs that inhibit phagocytosis, such as actin

polymerization inhibitors, Cytochalasin D. This would strongly support that internalization of particulate antigens is a phagocytic process. In D, antigen presentation of E α peptide on MHC II is shown by flow cytometry analysis and quantified. The presentation/proliferation assay is convincing.

Figure 2: Phagocytosis of particulate antigen by follicular B cells in vivo. The image shown in figure 2A is not very convincing. It shows the presence of only 1 μ m bead in the follicular B cell compartment. Is there an increase in the amount of beads that reach this area if spleens are analyzed after longer periods post-immunization?

For figure 2B, analogous to the observations for figure 1, how do the authors rule out that the anti-OVA staining differences (out vs in) observed in WT vs RhoG^{-/-} cells isolated from spleen do not correspond to changes in the degradative capacity of these cells and not their uptake by phagocytosis?

In 2D, the authors examine the capacity of phagocytosis in different subsets of B cells: MZ and FO cells. They show that both cell types are able to phagocytose particulate antigen in vivo to a similar extent. However, the authors do not comment on the result showing that the phagocytic capacity of MZ B cells is unaffected when RhoG is knocked out. This should be mentioned in the text.

Figure 4: Here the authors study RhoG dependent phagocytosis of alum-antigen aggregates by follicular B cells and the associated humoral response. Panel A shows an image of a follicular B cell interacting with alum-NIP aggregates, where a phagocytic cup appears. Does this structure appear in RhoG^{-/-} B cells? An image of these cells should be included, which would emphasize that these cells indeed display a defect their phagocytic capacity and consequently are unable to mount efficient humoral responses.

The data presented in panel C and D show that RhoG^{-/-} mice have impaired GC production and generation of high-affinity antibodies to particulate antigen, either in conditions measured in the entire mouse or when WT or RhoG^{-/-} B cells are adoptively transferred, thereby strongly suggesting that this defect is due to a defective phagocytic capacity of B cells and not of other cell types. Figure 4A shows a follicular B cell interacting with particulate antigen of at least 3 μ m in diameter, however the phagocytic capacity of these cells is 3 times lower when 3 μ m beads are used compared to 1 μ m (figure 1C). Are alum-antigen aggregates efficiently phagocytosed? This question should be addressed in the manuscript.

An important issue that the authors should consider and which would strengthen this report, is whether one can mimic the alum-based particulate antigen simply by immunizing with larger beads (2-3 μ m) containing antigens (NIP-OVA). The authors should include functional assays with larger beads.

Minor points

Figure 1A: scale bar is missing

Figure 1D: The graph on the right is missing a title on the y-axis.

Figure 2A should specify MOMA labeling for clarity.

1st Revision - authors' response

17 May 2018

Point-by-Point Response to reviewers

Reviewer 1 (R1) stated:

Secondary immune responses with high affinity antibodies rely on germinal center reactions during which follicular B cells interact with T follicular helper cells to acquire somatic mutations in their gene segments that encode the variable domains of Ig heavy and Ig light chains. This manuscript provides solid evidence for three major findings. First, follicular B cells are capable of capturing large particulate antigen complexes. Second, this process is initiated by cognate BCR antigen recognition and requires the small GTPase RhoG. Finally, this kind of antigen uptake drives affinity maturation and Ig class-switching. The presented in vitro and in vivo experiments fully support the authors' conclusion. Also, the manuscript is well written. Collectively, the results significantly contribute to our understanding of potent humoral immunity. Future experiments are necessary to elucidate the underlying signal mechanism in a more comprehensive manner. At this stage,

however, I do not miss additional experiments that will push the manuscript to another level. Hence, I recommend publication as is (which I rarely do).

We thank R1 for her/his positive comments and for encouraged us to continue this project to elucidate the signal mechanism in a more comprehensive manner.

Reviewer 2 (R2) stated:

This is an interesting paper demonstrating a new role for Rhog in antibody responses. Previously, no significant role for Rhog in antibody responses was found upon immunisation with soluble NP-KLH protein. However, the authors show here that B cell expression of Rhog is important in responses to particulate antigens, including antigens precipitated in alum. In the absence of Rhog in B cells, responses to particulate antigens are impaired in B cell germinal centre formation, class switching, and early production of high affinity antibodies. The authors explain this by showing that antigen-coated particles of 1-3 microns are phagocytosed by B cells and that Rhog is required for the phagocytic uptake and for efficient presentation of the antigens associated with the particles.

Previously, it was thought that uptake of particles that are too big for endocytosis (>200 nm) cannot be taken up by most B cells, except for specialised B cell subsets. However, some of these particles did not contain antigens specific for the BCR. In contrast, there were previous reports showing that the BCR can mediate phagocytic uptake of antigen-coated particles. The data here is novel in that they clarify the role of BCR-driven phagocytosis in B cells and identify Rhog as a component of this pathway that is important for antigen uptake and presentation.

The data is mostly of high quality and in my opinion support the main conclusions. I have however questions about some of the experimental details and also on how some of the data is interpreted.

We thank R2 for highlighting the novelty of our work and the high quality of our data. We also thank R2 for her/his feedback on how to improve our manuscript that is addressed bellow.

1.- R2 stated: *The phagocytosis by B cells is a bit misleadingly analysed in the literature, particularly with respect of the particles that are used for the experiments (latex beads coated or uncoated, bacteria etc). This lead to the early conclusions that only B1 B cells or B cells in certain species can phagocytose. I suggest a more comprehensive discussion and comparison of the published experiments to explain why the author's experiments differ from others. The authors should also cite Souwer et al, J. Immunol, 2009, who show BCR-mediated phagocytosis in Ramos B cells, which agrees with their data that B cells are capable of phagocytosis of antigen-coated particles.*

We thank and agree with R1 for pointing this out. We have now discuss this issue better in the text (page 12, lines 326-339), We also apologize for the missing reference in our original manuscript. The referred publication has now been cited in the revised manuscript.

2.- R2 stated: *In Fig. 1, the phagocytic index needs better explanation. The provided formula seems to sum cell numbers with phagocytosed beads, which does not fit with the numbers in the graphs. Are the numbers normalized to total numbers of B cells with beads?*

We apologize for not make clear enough the concept of phagocytic index. It is clear that our original description led to some confusion. We think that we have now clarified the concept of phagocytic index in the new manuscript version (page 5, lines 117-119).

3.- R2 stated: *In Fig 1D the legend mentions Rhog^{-/-} cells, but the figure only shows one set of data, presumably WT.*

We thank R2 for pointing this out. We have changed the figure legend accordingly (Fig 1E, line 766).

4.- R2 stated: *In Fig 1E, it would be important to know that the absence Rhog does not affect general antigen presentation to T cells, for example using a soluble peptide control.*

We thank R2 for this suggestion. We have now performed a new experiment addressing R2's comment which is shown in Supplementary Fig. 2 and discussed in the revised text (pages 6-7, lines 166-173). Using B1-8^{hi} system, we show that the absence of RhoG affect only the presentation of antigen to OT2 T cells when the antigen was given bound to beads but not in solution. These results further support the idea that follicular B cells phagocytose particulate antigens by a RhoG-dependent process.

5.- R2 stated: Fig 3A shows presence of beads in follicles, but not in the GC. Is this a reproducible finding? If yes, it suggests that GC B cells do not acquire the antigen from the beads, but from other sources. It would be interesting if the authors could discuss what the source of the antigen is. This could be some soluble antigen contamination in the immunisation prep, or release of antigen from the beads in vivo. The release can happen by extracellular degradation, or perhaps after phagocytosis by B cells or other cells. This is important for the interpretation of the GC experiments, and for effects on antibody production, as it suggests that Rhog dependent phagocytosis is important only during the initiation of the response. Such a conclusion is supported by the delay, rather than absence, of antibody production in Rhog-/- mice in the subsequent experiments. Also, the text indicates the analysis is 5 days post immunisation, the figure indicates day 7.

Although the number of GC B cells is much lower than the number of B cells in follicles, what we show in Figure 3A seems to be a tendency. The image is representative of other 18 GCs analyzed from 6 immunized mice. Our interpretation, such as the Reviewer suggests, that the GCs found at day 5 resulted from phagocytic events that took place much earlier. The B cells that phagocytosed soon after bead inoculation would have proliferated and diluted the phagocytosed beads among their progeny. So, it would be unlikely that we detect GC B cells with beads that initiated the GC reaction. In order to determine if the GC B cells present at day 5 after inoculation can or cannot phagocytose beads we would have to re-inoculate mice at that time point. We have tried to briefly discuss this in our revised manuscript (page 8, lines 216-219).

We apologize for the confusion regarding number of days after immunization in Fig. 3. Analysis was done after 5 days. This has been now changed in the figure legend.

6.-R2 states: 6. In Fig 4B, the authors measure separately high and low affinity antibodies by using low and high valency of the antigen, but technically, the multivalent antigen is going to bind both low and high affinity antibodies. A separate measure of "avidity" determined from the ratio of the two measurements is better. What day post immunisation were the IgM antibodies measured?

We thank R2 for this suggestion. We have included now a new graph showing the ratio between high and low affinity antibodies (Fig. 5A) and we have commented about these new observation in the text (page 10, line 279).

The IgM antibodies were measured after 14 days, we have made this clearer in the text and figure legend (page 10, line 276).

7.- R2 states: Also in Fig 4, the overall titers of the IgG1 antibodies are different between the two antigen formulation. This should be mentioned in the interpretation.

We appreciate R2's observation. We have now mentioned these differences between the different antigen formulations in the revised the text (page 9, lines 270-273).

8.- R2 states: In Supplementary Fig 3B,C, the authors conclude that the Rhog-/- cells respond normally to IgM and CpG stimulation. The more relevant stimulus, however, is particulate antigen. When the B cells cannot phagocytose the antigen coated beads, do they proliferate or survive normally? This is important to judge whether BCR signalling or other cell physiology in Rhog-/- B cells is normal in the context of particulate antigens.

We thank R2 for pointing this out and we do agree with R2 that this is a very important aspect to evaluate and to take into account to understand the role of RhoG. Consequently, we have now performed the requested experiment and the results are shown in Supplementary Fig. 4B and 4C. Importantly we have seen that the proliferative response of B cells to bead-bound anti-IgM is

impaired in the absence of RhoG, suggesting that RhoG not only controls the phagocytosis of particulate antigens, also controls BCR signaling. These findings further support our *in vivo* observations, but further work will be required to describe in more detail the role of RhoG in BCR signaling after particulate antigen phagocytosis. All this new set of data is now discussed in the revised manuscript (page 10, lines 290-293).

9.- R2 states: In the discussion, the authors should refrain from concluding that they "prove that phagocytic activity of B cells is key to explain the potent adjuvant effect of alum". The presented data is not sufficient to support this conclusion. Alum does many things and the data in this manuscript do not show how the stimulation of B cell phagocytosis compares to other effects of alum. The data only show that for antigen mixed with alum, B cell expression of Rhog, and therefore B cell phagocytosis, is important.

We thank R2 for this comment. We do agree with the R2 about the fact that Alum does many other things, in addition to promote antigen phagocytosis by B cells, to exert its potent adjuvant effect. We also think that R2 is right about the fact that although we have proved that the differences are not due to TLR signaling, there are other things apart from phagocytosis that could be also mediated by Alum. We have softened our text in the discussion of the new manuscript according to R2 suggestion (page 12, lines 318-319; page 14, lines 409-412).

10.- R2 states: In the abstract and discussion, a range of 1-10 microns is mentioned as a desirable size of particles for immunisation. However, the B cells could not phagocytose the 10 micron beads. 1-3 microns is a more realistic range.

We thank R2 for pointing this out. We have changed this range in the abstract and discussion.

Reviewer 3 (R3) stated:

*In this manuscript the authors address an interesting and relevant topic, the mechanisms used by B cells to internalize large particulate antigens and its importance to generate efficient humoral responses. In particular, they show that follicular B cells are able to phagocytose large antigen-coated particles through their BCR and that this is a fundamental property, which enables efficient humoral responses to such antigens. The authors provide *in vitro* and *in vivo* evidence that BCR-mediated phagocytosis by follicular B cells exists and that this process is dependent on the GTPase RhoG. The role of BCR-dependent phagocytosis in humoral responses is further supported by results showing that RhoG knockout mice have defective humoral responses to particulate antigens (1 μ m), whereas responses elicited by soluble antigens do not seem to be affected. Altogether, these findings are very interesting, however the idea that the BCR mediates phagocytosis of particulate antigens is not original, *per se*, and has been previously addressed by Batista and Neuberger (Embo J, 2000). The novelty of this work resides on the role of RhoG in mediating phagocytosis of particulate antigens by B cells and its role in promoting humoral responses *in vivo*. The results presented here are appealing to both the communities in cell biology and immunology. Overall, I recommend this paper for publication in this journal, however I feel certain points should be addressed:*

We thank R3 for considering the topic interesting and relevant not only for the immunology community also for cell biology community. We also appreciate the generous remarks on the current study and the relevant comments raised. The points raised by R3 have been addressed as follows:

1.- R3 states: Figure 1: Follicular B cells are able to phagocytose particulate antigens. The authors show 2 experimental approaches to demonstrate the phagocytic capacity of follicular B cells. In panel A, immunofluorescent labeling of actin (phalloidin) and anti-IgG (bead) are shown. Beads that are negative for anti-IgG staining are considered to be completely internalized and thus are "protected" from labeling. How do the authors distinguish between antigen degradation on the bead, which would also give negative anti-IgG staining and phagocytosis? The authors should include a set of permeabilized cells to show that the beads remain positive for IgG, under these conditions. Although phagocytosed beads are clearly distinguished from those that are only bound to the cell surface, the experiment would also benefit by staining plasma membrane (and not just actin), where engulfment of beads could be better appreciated.

We thank R3's comments and suggestions. We do understand R3's concerns about anti-IgG staining; we have therefore performed new experiments taking into account R3's suggestions. New results are now shown in Supplementary Fig 1B and Fig. 1D. We think that the new staining using anti-B220 (as recommended) and the controls probing that beads are protected from labelling due to their intracellular localization and not to antigen degradation, make clearer our claim. We have commented these new results in the revised manuscript (page 6, lines 139-148).

2.- R3 states: Figure 1: The data presented suggests that phagocytosis relies on engagement of the BCR and RhoG, however this report should include more information to address this point and not only functional readouts of the antibody responses. For instance, regarding the images, do the authors observe capping of the BCR in B cells interacting with antigen-coated beads? Do RhoG^{-/-} B cells show only surface-bound beads and how is actin organization affected in these cells? Images addressing these questions should be presented in this panel.

We welcome R3 comments and following her/his advice, we have performed a new staining using phalloidin to characterized better the actin cytoskeleton organization in *RhoG^{-/-}* cells (Supplementary Fig. 1D). We have discussed this result in the new manuscript (page 6, lines 143-148).

3.- R3 states: Figure 1: Next, the phagocytic process is quantified by flow cytometry (B-C). The results are convincing, however the authors should include the effect of drugs that inhibit phagocytosis, such as actin polymerization inhibitors, Cytochalasin D. This would strongly support that internalization of particulate antigens is a phagocytic process. In D, antigen presentation of Ea peptide on MHC II is shown by flow cytometry analysis and quantified. The presentation/proliferation assay is convincing.

We thank R3 positive comments regarding our convincing results about the phagocytic process quantified by flow cytometry and about antigen presentation. Using flow cytometry again, and following R3 suggestions we have evaluated the role of actin polymerization using Cytochalasin D and Latrunculin A, both well-known actin polymerization inhibitors. We show now that actin polymerization is key for the phagocytic process in B cells. Additionally, we have also used PP2 inhibitor to prove the dependence on BCR signaling. These new results are shown in Supplementary Fig. 1C. We have commented about these results in the revised manuscript (page 5, lines 128-131).

4.- R3 states: Figure 2: Phagocytosis of particulate antigen by follicular B cells in vivo. The image shown in figure 2A is not very convincing. It shows the presence of only 1µm bead in the follicular B cell compartment. Is there an increase in the amount of beads that reach this area if spleens are analyzed after longer periods post-immunization?

We thank R3 for pointing this out which indicates to us that we have not chosen the best figure to illustrate our results and conclusions. The Fig 2A in our former manuscript was a zoom in to illustrate a specific area of the tissue. We have now included in the new Fig. 2 (please see Fig. 2A) a panel with less amplification. In this figure is clearer the presence of numerous beads in the spleen at that time point. We hope, this new figure, together with the flow cytometry quantification (Fig. 2B), make our results more robust and convincing. We have discussed this in the new version of the manuscript (page 7, lines 183-186).

Regarding longer times, we have analyzed data at 5 hours and at 5 days after beads inoculation. We see a slight increase in the total number of beads (associated to total cells) at the longer time but the error bars are too big as to allow to draw a conclusion (see plot below). We think it is not worth to include this data in the paper.

5.- R3 states: Figure 2: For figure 2B, analogous to the observations for figure 1, how do the authors rule out that the anti-OVA staining differences (out vs in) observed in WT vs RhoG^{-/-} cells isolated from spleen do not correspond to changes in the degradative capacity of these cells and not their uptake by phagocytosis?

As before (please, see answer 1 to R3) we thank R3's concern. We hope that the new controls included following her/his suggestions (please see Supplementary Fig 1B), support our hypothesis that the differences between WT and RhoG^{-/-} are due to differences in number of beads in/out and not to changes in the degradative capacity of these cells.

6.- R3 states: Figure 2: In 2D, the authors examine the capacity of phagocytosis in different subsets of B cells: MZ and FO cells. They show that both cell types are able to phagocytose particulate antigen in vivo to a similar extent. However, the authors do not comment on the result showing that the phagocytic capacity of MZ B cells is unaffected when RhoG is knocked out. This should be mentioned in the text.

We thank R3's comment regarding these results. Is clear to us that we have not done a good job explaining our results in this part of the text. We have included a more extended description of the differences observed in vivo regarding follicular (FO) and marginal zone (MZ) B cells. As the Fig. 2C shows, FO and MZ B cells present similar phagocytic activity in WT cells, but the absence of RhoG affect to *both* cell subtypes. We have now mentioned this in the revised manuscript (page 7, lines 200-201).

7.- R3 states: Figure 4: Here the authors study RhoG dependent phagocytosis of alum-antigen aggregates by follicular B cells and the associated humoral response. Panel A shows an image of a follicular B cell interacting with alum-NIP aggregates, where a phagocytic cup appears. Does this structure appear in RhoG^{-/-} B cells? An image of these cells should be included, which would emphasize that these cells indeed display a defect their phagocytic capacity and consequently are unable to mount efficient humoral responses.

We thank R3 for this suggestion. We have performed a new experiment to visualize how RhoG^{-/-} cells interact with alum-NIP aggregates, which fail to internalize alum-NIP aggregates and to form a phagocytic cup. Results are now shown in Fig. 4B, and these new results are commented on the revised manuscript (page 9, lines 260-264).

8.- R3 states: Figure 4: The data presented in panel C and D show that RhoG^{-/-} mice have impaired GC production and generation of high-affinity antibodies to particulate antigen, either in

conditions measured in the entire mouse or when WT or *RhoG*^{-/-} B cells are adoptively transferred, thereby strongly suggesting that this defect is due to a defective phagocytic capacity of B cells and not of other cell types. Figure 4A shows a follicular B cell interacting with particulate antigen of at least 3 μm in diameter, however the phagocytic capacity of these cells is 3 times lower when 3 μm beads are used compared to 1 μm (figure 1C). Are alum-antigen aggregates efficiently phagocytosed? This question should be addressed in the manuscript.

We thank R3 for this suggestion because it encouraged us to develop a new technique to measure phagocytosis of alum-NIP aggregates, which reinforces our results and help us to quantify the defects previously shown by confocal microscopy (Fig. 4B and C). With this technique we corroborate that *RhoG*^{-/-} B cells present a defective phagocytic capacity when alum-antigen aggregates are used as immunogens. We have commented all these results in the new version of the manuscript (page 9, lines 260-264).

9.- R3 states: *Figure 4: An important issue that the authors should consider and which would strengthen this report, is whether one can mimic the alum-based particulate antigen simply by immunizing with larger beads (2-3 μm) containing antigens (NIP-OVA). The authors should include functional assays with larger beads.*

As suggested R3, we have tried to mimic the alum-based particulate antigen immunizing with larger beads (3 μm) coated with NIP-OVA. In this regard, we have performed a new and more extensive analysis of the response against beads of 1 and 3 μm (please, see Fig 3C, D and E). As expected, we have observed that *RhoG*^{-/-} mice presented a reduced percentage of germinal center B cells either with 1 or 3 μm beads. Interestingly, although with both sizes of beads antibody titers are reduced in *RhoG*^{-/-} mice, only using 3 μm beads significant differences were observed. This can be due, as we discussed in the revised manuscript (page 8-9), to a more stringent conditions in terms of *RhoG* requirement for the phagocytosis of bigger particles.

Minor comments:

Figure 1A: scale bar is missing.

Scale bars have been added

Figure 1D: The graph on the right is missing a title on the y-axis.

We have now provided a title

Figure 2A should specify MOMA labeling for clarity

MOMA labeling is specified in Figures 2A and 2B by the color code

2nd Editorial Decision

5 June 2018

Thank you for the submission of your revised manuscript to our editorial offices. We have now received the reports from the two referees that were asked to re-evaluate your study (you will find enclosed below).

As you will see, both referees now support the publication of your manuscript in EMBO reports. Before we can proceed with formal acceptance, I have the following editorial requests that we ask you to address in a final revised version of the manuscript.

- Please change the title to:

Antigen phagocytosis by B cells is required for a potent humoral response

-The manuscript is currently categorised as Scientific Report, which also fits to the number of displayed items (5 main figures and 4EV figures), however, the main manuscript text (introduction, results and discussion) is currently too long (nearly 35.000 characters including spaces). See:

<http://embor.embopress.org/authorguide#researcharticleguide>

We would therefore ask you to shorten this part of the manuscript (excluding material and methods, the references and the legends) to around around 28.000 characters (not more than 30000). To accomplish this, please also combine the Results and Discussion sections to one paragraph (termed Results & Discussion), removing any redundant parts.

- Please format the references according to our journal style. If there are more than 10 authors, 'et al' should be used, but keeping the first 10 authors. See:
<http://embor.embopress.org/authorguide#referencesformat>

- Please use the nomenclature 'Figure EV#' throughout the manuscript (not 'Supplementary Figure #').

Finally, please find attached a word file of the manuscript text (provided by our publisher) with changes we ask you to include in your final manuscript text, and some queries (comments), we ask you to address. Please provide your final manuscript file with track changes, in order that we can see the modifications done.

In addition I would need from you:

- a short, two-sentence summary of the manuscript
- two to three bullet points highlighting the key findings of your study
- a schematic summary figure (in jpeg or tiff format with the exact width of 550 pixels and a height of about 400 pixels) that can be used as a visual synopsis on our website.

REFEREE REPORTS

Referee #2:

I am satisfied with the response to my comments. I recommend the revised manuscript for publication.

Referee #3:

I have revised the new version of the manuscript and corresponding figures. The authors have responded to all of my concerns and have included the appropriate experiments/figures regarding bead phagocytosis and actin cytoskeleton dynamics in RhoG KO cells. I therefore recommend this paper for publication.

Corresponding Author Name: Balbino Alarcon

Manuscript Number: EMBOR-2018-46016V2